# Observation of Surface Displacement Associated with Rapid Urbanization and Land Creation in Lanzhou, Loess Plateau of China with Sentinel-1 SAR Imagery

**Yuming Wei [1], Xiaojie Liu [2,3], Chaoying Zhao [2,4,*] , Roberto Tomás [3] and Zhuo Jiang [2]**

[1] School of Civil Engineering, Lanzhou University of Technology, Lanzhou 730050, China; weiym1974@lut.edu.cn
[2] School of Geological Engineering and Geomatics, Chang'an University, Xi'an 710054, China; 2018026010@chd.edu.cn (X.L.); 2019126032@chd.edu.cn (Z.J.)
[3] Department of Civil Engineering, University of Alicante, 03080 Alicante, Spain; roberto.tomas@ua.es
[4] Key Laboratory of Western China's Mineral Resources and Geological Engineering, Ministry of Education, Xi'an 710054, China
[*] Correspondence: cyzhao@chd.edu.cn; Tel.: +86-29-8233-9261

**Abstract:** Lanzhou is one of the cities with the higher number of civil engineering projects for mountain excavation and city construction (MECC) on the China's Loess Plateau. As a result, the city is suffering from severe surface displacement, which is posing an increasing threat to the safety of the buildings. However, up to date, there is no comprehensive and high-precision displacement map to characterize the spatiotemporal surface displacement patterns in the city of Lanzhou. In this study, satellite-based observations, including optical remote sensing and synthetic aperture radar (SAR) sensing, were jointly used to characterize the landscape and topography changes in Lanzhou between 1997 and 2020 and investigate the spatiotemporal patterns of the surface displacement associated with the large-scale MECC projects from 2015 December to March 2021. First, we retrieved the landscape changes in Lanzhou during the last 23 years using multi-temporal optical remote sensing images. Results illustrate that the landscape in local areas of Lanzhou has been dramatically changed as a result of the large-scale MECC projects and rapid urbanization. Then, we optimized the ordinary time series InSAR processing procedure by a "dynamic estimation of digital elevation model (DEM) errors" step added before displacement inversion to avoid the false displacement signals caused by DEM errors. The DEM errors and the high-precision surface displacement maps between December 2015 and March 2021 were calculated with 124 ascending and 122 descending Sentinel-1 SAR images. By combining estimated DEM errors and optical images, we detected and mapped historical MECC areas in the study area since 2000, retrieved the excavated and filling areas of the MECC projects, and evaluated their areas and volumes as well as the thickness of the filling loess. Results demonstrated that the area and volume of the excavated regions were basically equal to that of the filling regions, and the maximum thickness of the filling loess was greater than 90 m. Significant non-uniform surface displacements were observed in the filling regions of the MECC projects, with the maximum cumulative displacement lower than −40 cm. 2D displacement results revealed that surface displacement associated with the MECC project was dominated by settlements. From the correlation analysis between the displacement and the filling thickness, we found that the displacement magnitude was positively correlated with the thickness of the filling loess. This finding indicated that the compaction and consolidation process of the filling loess largely dominated the surface displacement. Our findings are of paramount importance for the urban planning and construction on the Loess Plateau region in which large-scale MECC projects are being developed.

**Keywords:** Loess Plateau; Lanzhou; mountain excavation and city construction; surface displacement; InSAR; Sentinel-1; DEM errors; remote sensing images

## 1. Introduction

Urbanization is the process through which cities grow and expand in the process of modernization [1,2]. However, the lack of land is sometimes a significant impediment to rapid urbanization, particularly in the case of cities in mountainous areas, such as China's Loess Plateau [3]. As a consequence, a mountain excavation and city construction (MECC) project has become one of the major ways to solve the contradiction between more and more people and less and less land in the process of rapid urbanization. In cities such as Yan'an [4], Lanzhou [5], and Lanzhou New District [2], northwest China, tens of square kilometers of urban land have been created by large-scale MECC projects. The MECC projects can largely contribute to the development of local economy and urban construction. However, they can have a major impact on the local geological environment when the project intensity largely exceeds the geological bearing capacity, such as geohazards, soil erosion, as well as water and air pollution [3,4], among which the major concern is severe surface displacement. Severe surface displacement poses a significant threat to the construction of urban areas and may bring the risk of damaging the urban infrastructures, such as buildings, pipelines, and roads. Therefore, it is essential and of great significance to timely monitor and characterize surface displacements associated with MECC projects for facilitating a better understanding of its stability and spatiotemporal evolutions [6]. Furthermore, the derived outcomes can provide a valuable reference for subsequent land creation projects and urban construction and guarantee the safety of urban infrastructures to some extent.

In situ observation techniques, such as Global Navigation Satellite System (GNSS), leveling, and piezometers, can be used to monitor surface displacement with very high precision [7,8]; however, these techniques are usually limited in the application in wide area monitoring due to low spatial resolution [9]. As a result, it is challenging to retrieve the spatial pattern of surface displacement caused by the MECC project. Satellite-based interferometric synthetic aperture radar (InSAR) provides an excellent monitoring tool to map surface displacements with high spatiotemporal detail, i.e., spatial resolution ranging from sub-meter to dozens of meters, and centimeter to sub-centimeter measurement precision with weekly or monthly updates [10]. The strengths of this earth observation technique have been adequately illustrated in its extensive applications, such as reclamations [11,12], infrastructures [13,14], glaciers [15], earthquakes [16,17], volcanos [18,19], landslides [20,21], and nuclear explosion [22]. InSAR has also become a powerful tool to map and characterize settlements associated with the large-scale MECC projects. For instance, Chen et al. [2] detected land subsidence related to land creation and rapid urbanization in Lanzhou New District, Gansu province of China, using time series InSAR; Wu et al. [23] detected and mapped surface displacement caused by the mega-scale MECC project in Yan'an, Shaanxi province of China, using SBAS-InSAR method. However, those recent InSAR studies on surface displacement monitoring associated with MECC projects were mainly focused on the one-dimensional line-of-sight (LOS) direction or vertical direction (directly projected by assuming that no horizontal displacement occurred); this is satisfactory for detecting the spatial distribution of the active displacement areas, but it is insufficient for investigating the spatial displacement patterns of the created land in details. Furthermore, the ongoing MECC projects can lead to dramatic changes in local terrain [4], thus resulting in large digital elevation model (DEM) errors in SAR data processing. The time-variable pseudo displacement signals can be generated in the case when the DEM errors are not exactly corrected [24]. However, a completed processing procedure for the DEM errors is not obtained in those previous studies [2,4,5,23].

Lanzhou, Gansu province, is one of the significant central cities in western China, and it is also an important node city in the economic belt of the Silk Road. The city was developed in a fluvial valley area in China's Loess Plateau and is characterized by a limited available land for urban development. Consequently, large-scale MECC projects have been launched to solve the problem of lack of available land. According to the government's planning report [25], approximately 700 mountains have been excavated to construct urban

land by October 2012 in the city of Lanzhou. As a result, the MECC project in Lanzhou became one of the largest geotechnical challenges to create urban land in a collapsible loess area in the world. These unprecedented MECC projects have resulted in severe surface displacements in Lanzhou, some of which have threatened the safety of buildings [5], increasingly attracting attention of society and scientists. Several studies on the spatial distribution and mechanism of surface displacement have been conducted in this area, using InSAR methods such as those published by He et al. [5] and Li et al. [26]. These authors monitored and characterized the surface displacement in Lanzhou from March 2015 to May 2019 and from October 2014 to September 2016, respectively, using Sentinel-1 SAR images based on Small BAseline Subset (SBAS) and Persistent Scatterer (PS) InSAR methods. Wang et al. [27] analyzed the surface displacement and its driving force in Lanzhou by combining InSAR-derived displacement and geo-detector. However, up to date, a comprehensive, high-precision and detailed surface displacement measurement over the whole Lanzhou is absent. Furthermore, long-term temporal evolution of land after MECC projects, and the interrelations between surface displacement and MECC projects, are still not understood in depth, mainly due to the limited number of data exploited in previous studies [5,26,27]. Besides, there is no relevant literature to study how to map the area and volume of excavation and filling regions using SAR observations; this is very important for investigating the spatial displacement pattern of the created land in absence of high-resolution post-construction DEM data.

To this end, this paper presents a comprehensive spatial evolution analysis of land creation and high-precision surface displacement measurements in Lanzhou during the past 20 years, using satellite-based optical remote sensing and InSAR observations. This is the first time that the combination of multi-source SAR images, high-resolution optical images, and DEM from the shuttle radar topography mission (SRTM) is used to investigate the spatiotemporal surface displacement patterns and characteristics associated with large-scale MECC projects. A set of 124 ascending Sentinel-1 images covering the period of December 2015 to April 2021, 122 descending Sentinel-1 images spanning from December 2015 to April 2021, and 6 optical remote sensing images during 1997–2020 were collected and used in this study. Utilizing a combination of ascending and descending SAR images has these advantages: (1) it improves the accuracy of active displacement areas mapping, (2) cross-validates InSAR-derived displacement results, (3) retrieves two-dimensional displacement fields, and (4) facilitates a better understanding of surface displacement characteristics. We also used the estimated DEM errors from Sentinel-1 images to detect and map historical MECC areas to extract the excavation and filling regions and their volume and to investigate the response of surface displacement to the thickness of the filling loess. We showed that DEM error is a major error source in SAR processing in areas with large-scale MECC projects, although it can be used as a valuable information to assist the investigation of surface displacement patterns.

## 2. Study Area and Datasets

### 2.1. Study Area

Lanzhou is situated in Gansu province of the China's Loess Plateau (Figure 1a) [5]. It is one of the most important industrial cities in northwest China and is also a significant node city along the Belt and Road. The study area has an area of approximately 1631 km$^2$ consisting of six districts/counties: Xigu district, Anning district, Qilihe district, Chengguan district, Gaolan county, and Yuzhong county (Figure 1b). The topography of the study area is low in the northeast and high in the southwest [27], with an average attitude of 1520 m a.s.l. The study area belongs to the temperate arid and semi-arid climate, with an annual average temperature of 11.2 °C and an annual average precipitation of 327 mm. Geologically, Lanzhou is located in an area covered by Quaternary loess. The largest thickness of the loess deposits exceeds 300 m, and the average thickness is in the range of 50–100 m [28]. The plasticity properties of loess material provide favorable conditions for excavating mountains and constructing cities. On the other hand, the water

sensitivity, highly porosity, and joints and fissures of loess make it sensitive to seismic activities, rainfall, and human activities [29]. This makes Lanzhou an area prone to serious soil erosion and frequent occurrences of geohazards including landslides, collapses, debris flows, and sinkholes [5,30].

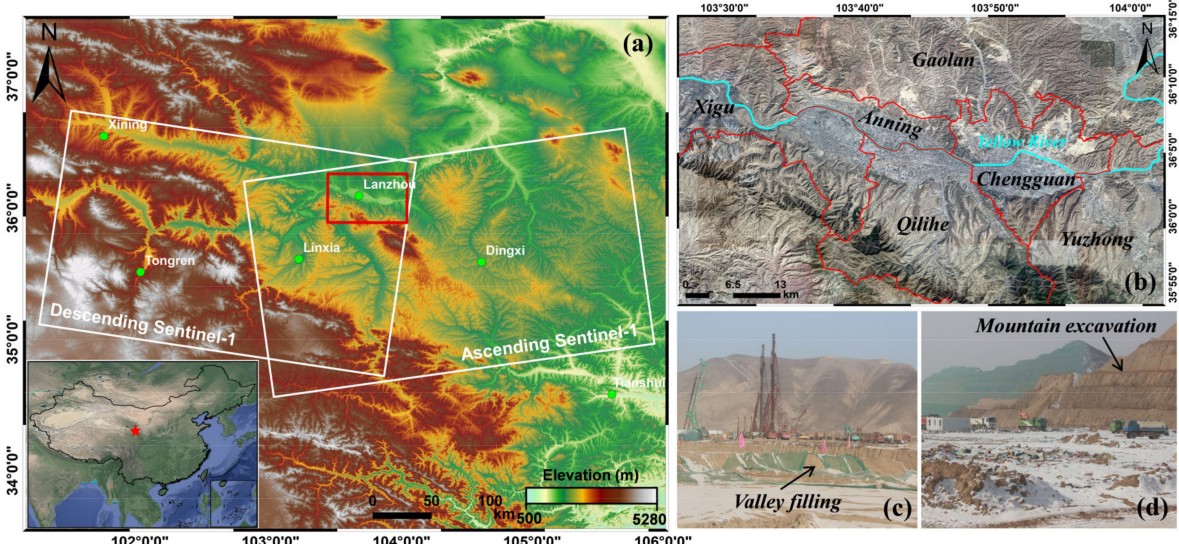

**Figure 1.** Overview of the study area: (**a**) location of Lanzhou (red star and red rectangle) as well as coverage of SAR images used in this study (white rectangles); (**b**) remote sensing image of the study area acquired in 2020, where red lines indicate the boundary of the county and the district; and scene photos of valley filling (**c**) and mountain excavation (**d**) in Lanzhou.

Lanzhou is a city that was developed on a series of river terraces of the Yellow River (Figure 1b), confined within a narrow east–west trending structural basin [31]. The mountains on the north and south sides of the city show steep slopes, high topographic relief and narrow valleys. However, the city has a relatively high population density, and population growth and industrial development have been very rapid in recent years. Consequently, the intrinsic topographical conditions of Lanzhou impose great restrictions on the space available for urban growth. Thus, the large-scale MECC projects were launched in 2000 to solve the issues of the increasing demand for new land for living and urban development. The MECC projects in Lanzhou can be categorized into four different stages [25], that is, the period between 2000 and 2005 is the beginning stage; the period from 2005 to 2010 is the accelerating stage; the period of 2010 to 2015 is the fast stage; and the period between 2015 and 2020 is the decelerating stage.

Figure 2 shows six optical remote sensing images of the city of Lanzhou between 1997 and 2020, and the blue polygons delineate the regions of ten large-scale MECC projects. It can be seen from Figure 2 that the number and areas of the MECC projects in Lanzhou have largely increased during the past 23 years. That is, that there is no region of the MECC projects in Lanzhou before 2000 shown in Figure 2a. However, the sporadic regions of the MECC projects can be observed around the main urban area of Lanzhou between 2000 and 2005, but their area is relatively small (Figure 2b,c). The number of the MECC projects increased significantly during the period of 2005 and 2010. It is worth noticing that the number of large-scale MECC projects was small in this period, and the largest land creation area is observed in Region C (Figure 2c,d). Evidence from Figure 2d,e shows that many large-scale MECC projects were launched during the period of 2010–2015, such as Regions B, E, G, and H, and the areas of land creation in this period were much larger than in other time periods. As seen from Figure 2e,f, the number of MECC projects continuously increased during the period of 2015–2020, and many buildings were constructed on the created land, such as Regions B, C, G, and H. For region B, multi-temporal optical images demonstrated that the MECC projects were launched after January 2011, and construction of the buildings started after March 2014. The excavation of mountains and land creation

in the Region C started after January 2001, it has already been basically completed, and a large number of buildings was constructed in December 2010. Among these MECC projects, the Region G, launched after July 2009, has the largest area. Up to date, the MECC projects in this region are still underway, whilst many buildings have been constructed on the created land.

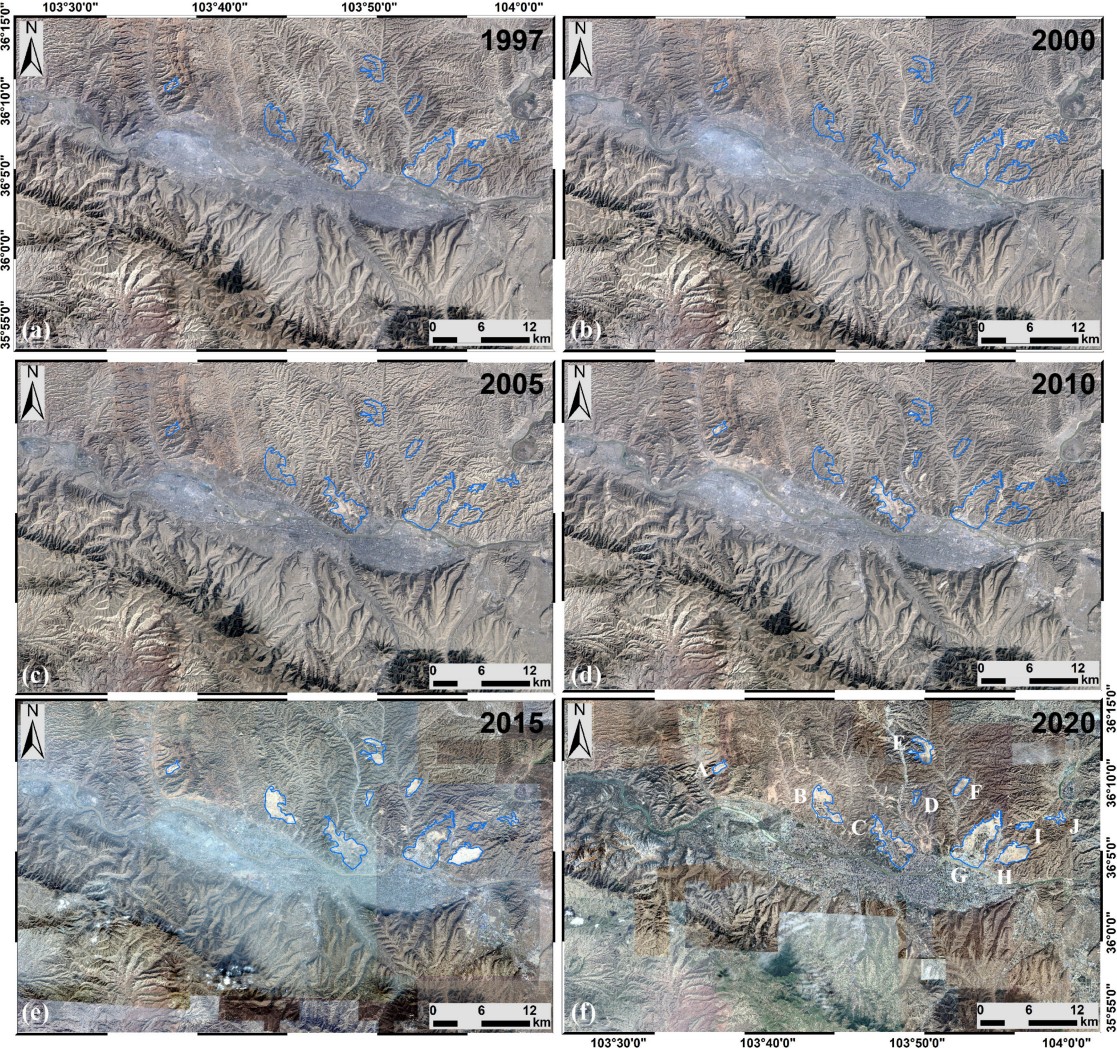

**Figure 2.** Optical remote sensing images of the city of Lanzhou at different time periods ((**a**)—in 1997, (**b**)—in 2000, (**c**)—in 2005, (**d**)—in 2010, (**e**)—in 2015, and (**f**)—in 2020), where the blue lines represent the regions of ten large-scale mountain excavation and land creation.

*2.2. Datasets*

2.2.1. SAR Datasets

Evidence from Figure 2 demonstrates that large-scale MECC projects were intensively performed in the study area during the period between 2000 and 2020, resulting in dramatic ground surface changes. Thus, in order to eliminate the influence of the decorrelation of SAR images caused by engineering activities and to obtain high-density InSAR monitoring points, as well as to comprehensively retrieve spatiotemporal displacement patterns of ground surface after MECC projects, a total of 246 SAR images—124 ascending and 122 descending—acquired from Sentinel-1 satellite covering the period of December 2015 to April 2021 were used. The spatial coverages of the used SAR images are shown in Figure 1a, and their detailed parameters are listed in Table 1. Sentinel-1 sensor operates at C-band (wavelength of 5.6 cm) with a large swath width of 250 km, providing the ground

resolutions of single-looking images in the range and azimuth directions of about 4 and 14 m, respectively. We selected the SAR images from ascending and descending tracks, for which the effects of the SAR geometrical distortions can be largely reduced, thus facilitating a more robust detection and mapping of the active displacement areas. Moreover, two-dimensional (2D) displacement maps of the areas of interest can be estimated using a combination of ascending and descending images, enabling a better understanding of surface displacement characteristics and patterns.

**Table 1.** Details of the datasets used in this study.

| Dataset | Optical Images | SRTM DEM | Sentinel-1 | Sentinel-1 |
|---|---|---|---|---|
| Orbit direction | - | - | ascending | descending |
| Heading (°) | - | - | −10.404 | −169.310 |
| Incidence angle (°) | - | - | 33.727 | 33.751 |
| Ground resolution | 1.2 m~19 m | 30 m | 4.20 (Rg) × 13.97 (Az) | 4.20 (Rg) × 13.97 (AZ) |
| Temporal coverage | 1997–2020 | 2000 | 12/2015–04/2021 | 12/2015–04/2021 |
| Number of images | 6 | 1 | 124 | 122 |

Note that Rg and Az stand for range and azimuth direction, respectively.

### 2.2.2. Optical Images and DEM

In order to analyze the spatiotemporal evolution of the MECC projects and assist in the detection of historical MECC areas, six optical remote sensing images from Google Earth Platform acquired in 1997, 2000, 2005, 2010, 2015, and 2020 were collected and exploited. These optical images, cloud-free and with clear ground features, have a spatial resolution ranging from 1.2 to 19 m (Table 1). Moreover, DEM data from the shuttle radar topography mission (SRTM) acquired in 2000, with a resolution of 30 m, were collected to remove the topographic phase component in SAR images and to estimate the DEM errors.

### 3. Methodology

As the local terrain has been dramatically changed, and it was still changing during the InSAR observation period, we adopted an optimized workflow presented in Figure 3, which consisted of the following six phases: (I) determination of the spatiotemporal evolution of the MECC projects, (II) dynamic estimation and correction of DEM errors, (III) estimation of one-dimensional (1D) displacement rates and time series in the line-of-sight (LOS) direction, (IV) calculation of two-dimensional (2D) displacement rates in the east–west and vertical directions, (V) detection and mapping of historical MECC areas, and (VI) retrieval of spatiotemporal surface displacement patterns and characteristics associated with MECC projects.

First, optical images and SAR intensity images were combined to map the boundaries of the MECC projects in the study area and to determine their spatiotemporal evolution by visual interpretation method. The main objectives of this phase were to assist (1) the dynamic estimation of DEM errors, (2) the detection of historical MECC areas, and (3) the investigation of surface displacement patterns. Second, based on the results obtained from Phase I, the whole SAR image stack for each track (i.e., ascending and descending) were divided into several sub-data stacks (i.e., Stack_1, Stack_2, . . . , Stack_n) according to the similarity of topographic features. The topographic features of all SAR images in each divided sub-data stack were basically consistent. The DEM errors for each sub-data stack were then estimated with the least square (LS) method and added to the original DEM to obtain the updated DEM stacks. The basic theory and process of the dynamic estimation of DEM errors are described in detail in Section 3.1. The unwrapped interferograms of each sub-data stack were then reconstructed using the updated DEM stacks. Third, the 1D displacement rate and time series of ascending and descending images in the LOS direction were estimated using the unwrapped interferograms produced in Phase II. The details of the 1D LOS displacement estimation are clarified in Section 3.2. Fourth, the 2D displacement rates in the horizontal east–west and vertical directions were calculated

by fusing the unwrapped interferograms from ascending and descending images. The method of the 2D displacement estimation is described in detail in Section 3.3. Next, we detected and mapped the historical MECC areas in the study area using the estimated DEM errors and the multi-temporal optical remote sensing images. Two main steps were involved to detect and map the historical MECC areas: (1) the preliminary detection of suspected MECC areas was conducted using the estimated DEM errors; i.e., it was regarded as a suspicious MECC area in the case that the absolute value of the estimated DEM error was greater than 10 m; (2) the suspicious MECC areas were superimposed onto the multi-temporal optical images and were further screened according to the landscape and topography changes, and the misjudged areas were removed to generate the final distributions of the historical MECC areas. Finally, the results generated in the previous four phases were combined to investigate the spatiotemporal displacement patterns and characteristics of ground surface in the study area, including the determination of the relationship between surface displacement and large-scale MECC projects via overlay analysis, the investigation of the response of surface displacement to the thickness of filling loess through correlation analysis, and the study of the long-term temporal evolution of surface displacement associated with the mountain excavation and valley filling projects using InSAR-derived displacement time series.

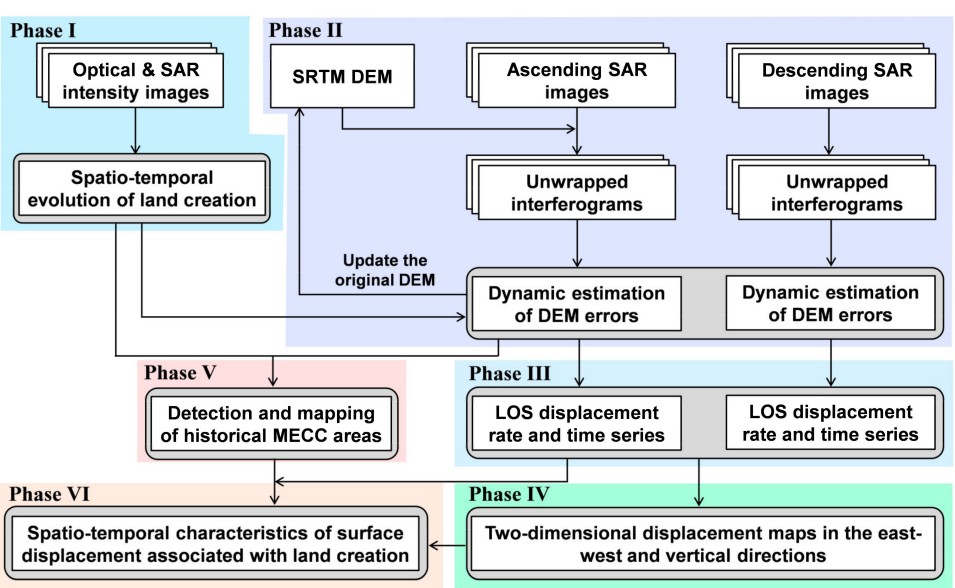

**Figure 3.** Workflow of the methodology adopted in this study.

### 3.1. Dynamic Estimation of DEM Errors

The large DEM errors in this study were caused by the large-scale MECC projects. The time-variable signals caused by DEM errors are easy to be misinterpreted as surface displacement [24]. In addition, the dense fringes in interferograms caused by DEM errors can increase the difficulty of phase unwrapping and even result in unwrapping errors [32]. Thus, accurate estimation and correction of DEM errors play an essential role in the retrieval of surface displacement in areas with large-scale MECC projects. Taking into account that $N + 1$ SAR images were acquired, and those generated $M$ unwrapped interferograms, the interferometric phase ($\Delta\phi_j$) for each unwrapped interferogram can be mathematically expressed by the following equation (Equation (1)) [33,34]:

$$\Delta\phi_j = \frac{4\pi}{\lambda} \cdot \Delta r_j + \frac{4\pi}{\lambda R \sin\theta} \cdot B_{\perp}^{j} \cdot \Delta h_j + \Delta n_j \tag{1}$$

where $\Delta r_j$ and $\Delta h_j$ stand for the surface displacement and the DEM error, respectively; $\lambda$, $R$, and $\theta$ are the wavelength of SAR images, distance of SAR sensor to ground target, and the incidence angle of SAR satellite, respectively; $B_{\perp}^{j}$ represents the perpendicular

baseline of the interferograms; and $\Delta n_j$ is composed of the residual phases associated with atmospheric artifacts, orbital errors and noise phases. For the $M$ unwrapped interferograms, Equation (1) can be expanded to the matrix format shown in Equation (2).

$$\Delta\phi_{M\times1} = \underbrace{\left[\begin{array}{cc} \frac{4\pi}{\lambda}\cdot\Delta r & \frac{4\pi}{\lambda R\sin\theta}\cdot B_\perp \end{array}\right]}_{A} _{M\times N} \cdot \left[\begin{array}{c} \Delta r \\ \Delta h \end{array}\right]_{N\times1} + \Delta n_{M\times1} \qquad (2)$$

where $A$ is the design matrix. Thus, the unknown parameter of DEM errors can be estimated by solving Equation (2) using the least square (LS) method, as shown in Equation (3).

$$\left[\begin{array}{c} \Delta r \\ \Delta h \end{array}\right] = (A^T\cdot A)^{-1}\cdot A^T\cdot\Delta\phi \qquad (3)$$

The MECC projects in local regions of the study area were still ongoing during the InSAR observation period of this study, thus causing dynamic changes in the terrain. Consequently, the workflow presented in Figure 4 was adopted to dynamically estimate DEM errors. First, the whole stack of SAR images was divided into several sub-data stacks based on the similarity of topographic features, and the unwrapped interferograms in each sub-stack were then generated independently. The detailed procedures for generating the unwrapped interferograms are presented in Section 3.2. It is worth to specify that the topographic phase in the interferograms in the DEM error estimation was estimated and removed using SRTM DEM. In the estimation of DEM errors, we divided the whole stack of SAR images into two sub-stacks. The first sub-stack included SAR images acquired between December 2015 and June 2020, and the second one included SAR images acquired between June 2020 and March 2021. Second, the high-quality interferograms characterized by small temporal baseline and large spatial baseline and atmospheric artifacts-free were selected in each sub-stack to estimate the initial DEM errors using Equations (2) and (3). In our SAR processing, the minimum spatial baseline ($B_\perp$) was set to 80 m, and the maximum temporal baseline ($B_T$) was set to 24 days. In addition, the low-coherence areas in the interferograms were masked to avoid false estimation. Third, the estimated initial DEM errors were filtered, interpolated, and geocoded to produce the final stacks of DEM errors. We used the median filtering to remove outliers in the estimated DEM errors, the inpaint_nans function [35] was used to interpolate the NaN elements caused by the low coherence, and the stacks of DEM errors were finally geocoded into the WGS 84 coordinate system for further analysis.

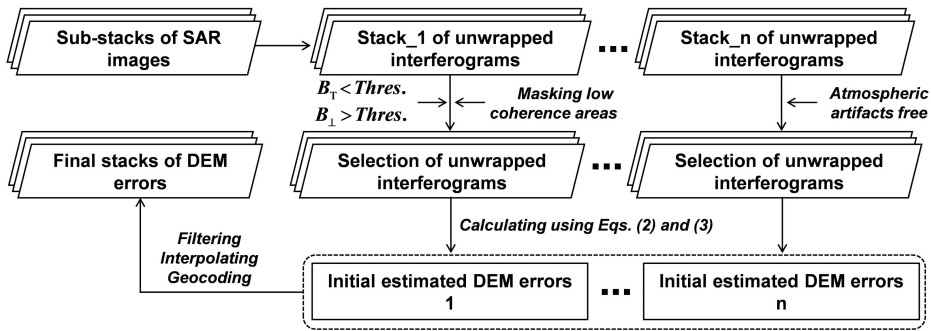

**Figure 4.** Flowchart of dynamic estimation of DEM errors, where Thres. indicates the threshold, and Eqs. represents the equations.

### 3.2. One-Dimensional LOS Displacement Estimation

LOS surface displacements of the study area from ascending and descending Sentinel-1 images were estimated using the SBAS-InSAR [33] approach. The SAR images were processed using GAMMA software [36]. Distributed MECC projects, rapid urbanization, and larger-gradient displacement jointly contributed to decorrelation noises in interferograms. Phase coherence can be effectively improved by SBAS-InSAR method by clustering

optimal subsets of coherent interferograms. First, an optimal primary image was selected by taking into account the temporal and spatial baselines and Doppler central frequency variations, and all the secondary images were accurately co-registered to the geometry of the selected primary image. Then, we generated all possible interferograms by setting the thresholds of temporal and spatial baselines among the co-registered SAR images. Only interferograms with small temporal (<100 days) and spatial (<200 m) baselines were generated to maintain good coherence and form a redundant observation network. The topographic phase was estimated using the update DEM stacks generated according to Section 3.1, and then it was subtracted from the generated interferograms. The influence of DEM errors on displacement estimation can be suppressed through the use of update DEM stacks. To reduce the speckle noise and to map small-scale surface displacement areas, the interferograms were multi-looked using a factor of $4 \times 1$ in the range and azimuth directions. Next, the interferograms were filtered using an adaptive filtering function based on the local fringe spectrum [37] and unwrapped using the approach of minimum cost flow [38]. The high-frequency noise signals could be removed by the filtering procedure, thus largely reducing the difficulty of phase unwrapping. As our study area is located in the plateau area, the stratified component of tropospheric phase delays could not be neglected. Thus, the stratified delays were corrected using an empirical linear relationship between the interferometric phase and the elevation [39]. In addition, we used a biquadratic model [6] to correct the artefacts of orbit errors and atmospheric disturbance. After correcting the errors and artefacts in the unwrapped interferograms, we conducted a careful manual selection of the interferograms. In this step, the interferograms with very low coherence were excluded, and those with high quality were finally selected to estimate the displacement maps. A total of 934 high-quality interferograms were selected and used finally in this study, including 520 interferometric pairs from ascending images, and 414 from descending images. Such a large number of observations can ensure the high-precision estimation of surface displacement. Figure 5 shows the spatiotemporal baseline information of the selected interferometric pairs. Finally, we used the method of the weighted averaging of interferograms [40] to estimate the displacement rates from ascending and descending unwrapped phase, and the displacement time series were calculated with the LS method.

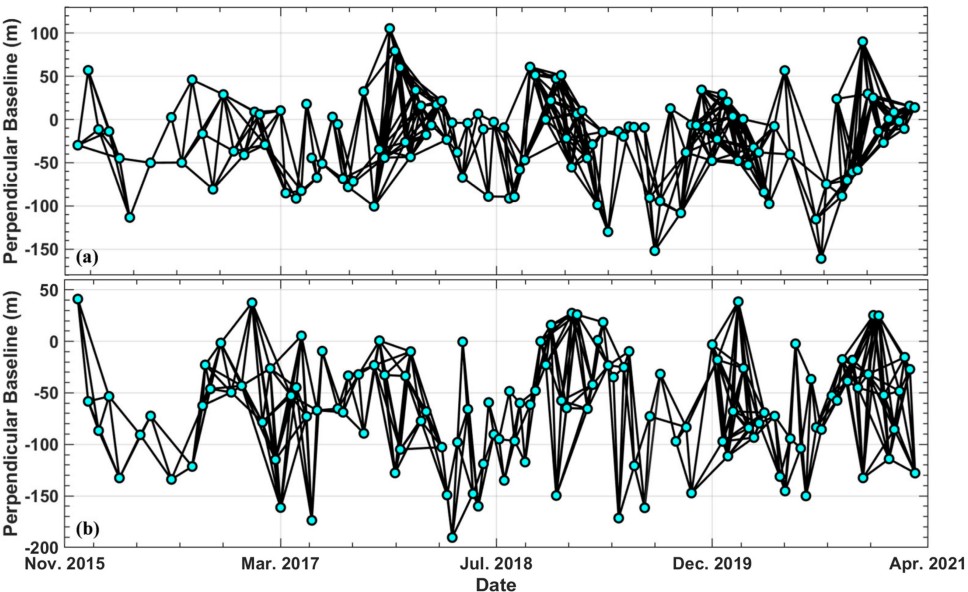

**Figure 5.** Spatiotemporal baseline information for ascending (**a**) and descending (**b**) Sentinel-1 images. The lines indicate the interferometric pairs, and the circles show SAR acquisitions.

### 3.3. Two-Dimensional Displacement Estimation

The horizontal east–west and vertical components of the surface displacement can be retrieved when ascending and descending images are available [41]. On the basis of

viewing geometry of SAR satellites, the relationship between the real three-dimensional (3D) surface displacement and the InSAR LOS measurements can be mathematically expressed as follows [42]:

$$
\begin{bmatrix} d_{los}^{asc} \\ d_{los}^{dsc} \end{bmatrix} = \begin{bmatrix} \sin\theta^{asc}\sin\alpha^{asc} & -\sin\theta^{asc}\cos\alpha^{asc} & \cos\theta^{asc} \\ \sin\theta^{dsc}\sin\alpha^{dsc} & -\sin\theta^{dsc}\cos\alpha^{dsc} & \cos\theta^{dsc} \end{bmatrix} \cdot \begin{bmatrix} v^N \\ v^E \\ v^U \end{bmatrix} \tag{4}
$$

where the superscripts *asc* and *dsc* stand for the ascending and descending images, respectively; $\theta$ and $\alpha$ correspond to the incidence angle and the flight direction angle of the SAR satellite, respectively; and $v^N$, $v^E$, and $v^U$ are the surface displacement rates in the north, east, and vertical directions, respectively. We can see from Table 1 that the incidence angles of ascending and descending Sentinel-1 images in the study area were basically the same; therefore, Equation (4) is an ill-posed inverse problem, where any small fluctuation in observations can result in a large change in the estimated displacement rates. In addition, InSAR measurements are extremely insensitive to the north–south displacement component, as are the near-polar orbiting characteristics of all SAR sensors. Thus, the Tikhonov regularization (TR) method [43] was introduced to increase the stability of the Equation (4), and the normal equation of 2D displacement inversion can be rewritten as Equation (5) by ignoring the north–south contribution [42].

$$
\begin{bmatrix} A \\ \lambda L \end{bmatrix} \cdot \begin{bmatrix} v_E \\ v_U \end{bmatrix} = \begin{bmatrix} d \\ 0 \end{bmatrix} \tag{5}
$$

where $A$ is the coefficient matrix; $\lambda$ indicates the regularization parameter, which can be determined via the L-curve method [42]; and $L$ stands for the order of the regularization. The 2D displacement rates $v_E$ and $v_U$ can be estimated by solving Equation (5) using the singular value decomposition (SVD) method.

## 4. Results and Analyses

### 4.1. Assessment of DEM Error Correction

Large DEM errors exist in interferograms with long spatial baseline due to the topographic changes caused by large-scale MECC projects. Thus, we applied the method described in Section 3.1 to correct potential DEM errors in the interferogram stack. Figure 6 shows four exemplary unwrapped interferograms from ascending images before (i.e., a and c) and after (i.e., b and d) correcting DEM errors, Figure 7 shows examples of descending images without (i.e., a and c) and with (i.e., b and d) DEM errors correction, and Figure 8 shows the displacement rate of the study area between December 2015 and December 2017 from descending Sentinel-1 images calculated without (a) and with (b) DEM errors correction. Spatial baselines of these exemplary interferograms were all greater than 120 m, and temporal baselines were all smaller than 24 days. We can see from Figure 6a,c and Figure 7a,c that the DEM errors in the study area were very conspicuous, particularly in the MECC areas, hampering the accurate mapping of the displacement pattern and can be easily misinterpreted as displacement signals. In contrast, after the correction of DEM errors, we can see from Figure 6b,d and Figure 7b,d that the phase components caused by DEM errors have been effectively removed. The accurate displacement field of ground surface can be identified using the corrected interferograms, benefiting the further estimation of displacement rates and time series. Furthermore, as can be seen in the displacement rate map calculated without DEM errors correction (Figure 8a), strong signals of uplift displacement were measured in the MECC areas (see black rectangles) with a magnitude of 26 mm/year. However, there is no obvious uplift displacement signals in the displacement rate map calculated with DEM errors correction. This fact demonstrates that uplift displacement signals were caused by the DEM errors, suggesting that the DEM errors should be carefully checked and corrected in the surface displacement mapping associated with large-scale MECC projects.

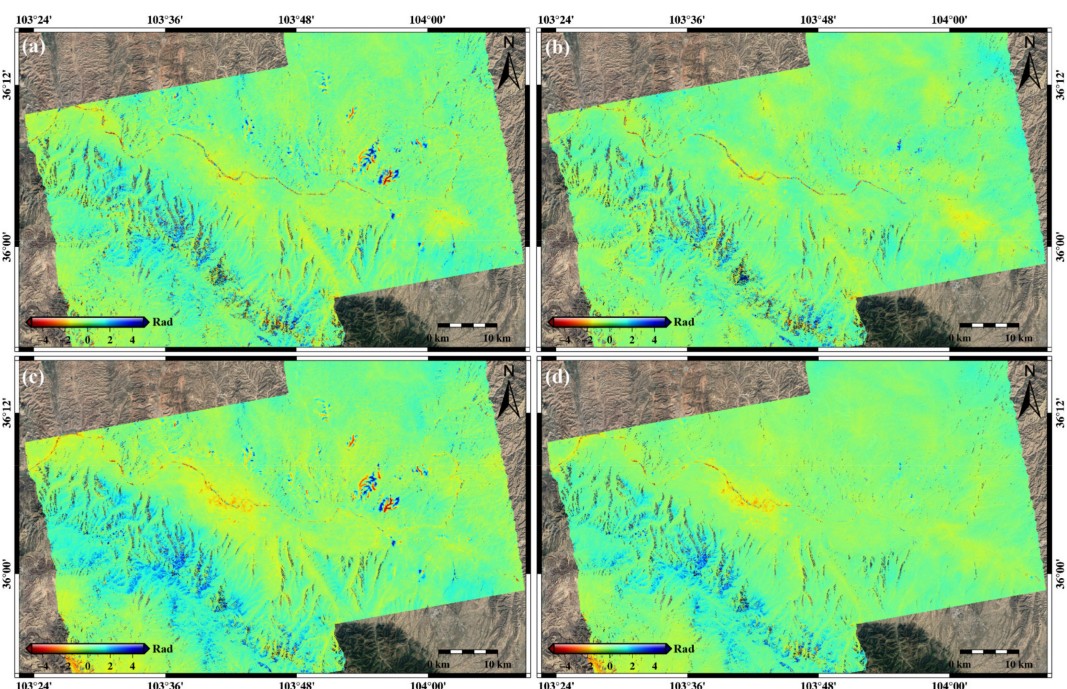

**Figure 6.** Exemplary unwrapped interferograms without (left) and with (right) DEM errors correction from ascending Sentinel-1 images: (**a**) interferogram generated between 6 November 2020 and 30 November 2020 before DEM errors correction, with spatial and temporal baselines of 150 m and 24 days, respectively; (**b**) the same interferogram after DEM errors correction; (**c**) interferogram generated between 18 November 2020 and 30 November 2020 before DEM errors correction, with spatial and temporal baselines of 148 m and 12 days, respectively; (**d**) the same interferograms after DEM errors correction.

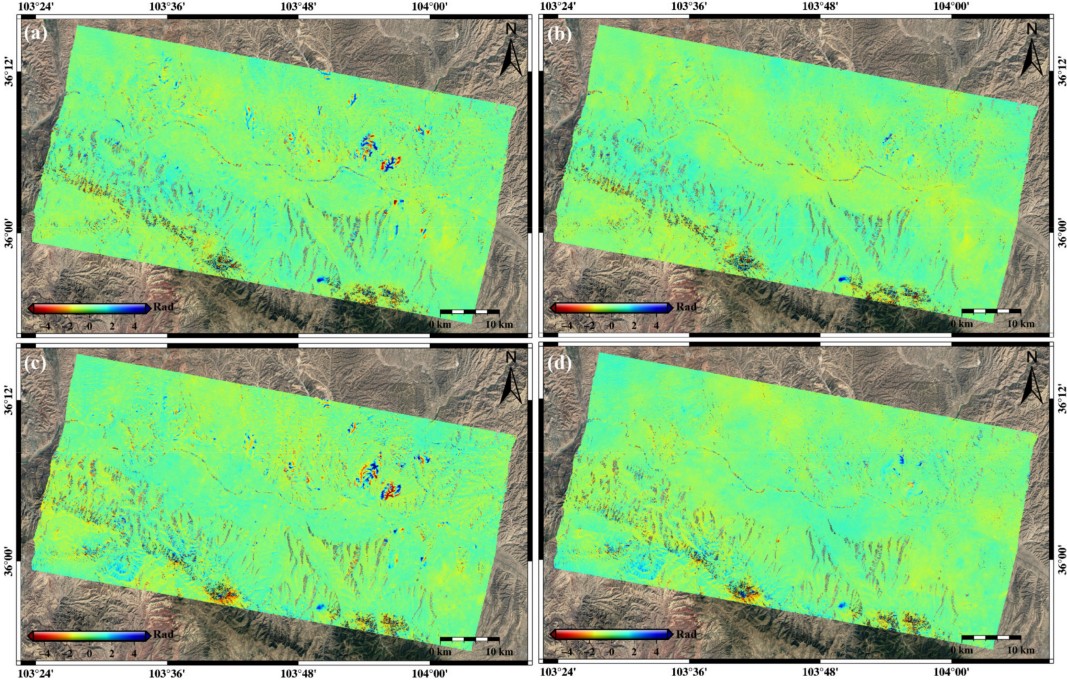

**Figure 7.** Exemplary unwrapped interferograms without (left) and with (right) DEM errors correction from descending Sentinel-1 images: (**a**) interferogram generated between 22 November 2018 and 16 December 2018 before DEM errors correction, with spatial and temporal baselines of −127 m and 24 days, respectively; (**b**) the same interferogram after DEM errors correction; (**c**) interferogram generated between 23 November 2020 and 17 December 2020 before DEM errors correction, with spatial and temporal baselines of 158 m and 24 days, respectively; (**d**) the same interferogram after DEM errors correction.

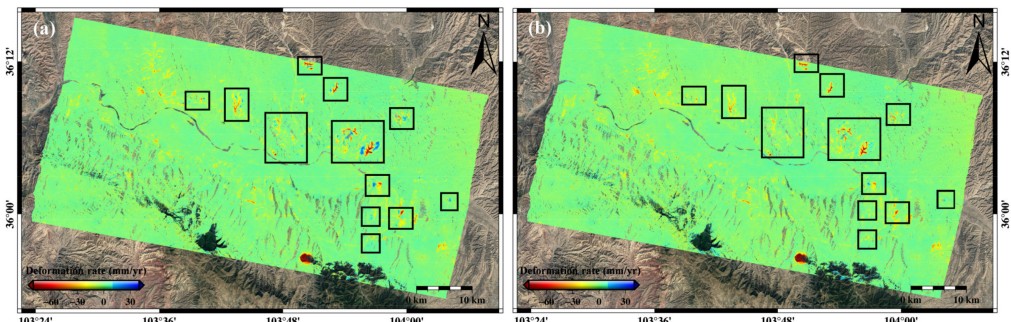

**Figure 8.** Displacement rates of the study area between December 2015 and December 2017 derived from descending Sentinel-1 images processed without (**a**) and with (**b**) DEM errors correction.

### 4.2. Detection and Mapping of Historical MECC Areas Using the Estimated DEM Errors

As can be seen from Section 4.1, false displacement signals caused by DEM errors existing in the interferograms can deceive the displacement interpretation. However, the estimated DEM errors can be used as a secondary product to detect and map historical MECC areas. Thus, in this paper, the method described in Section 3.1 was applied to obtain accurate estimations of DEM errors from ascending and descending images. Figure 9 shows the estimated DEM errors from ascending (Figure 9c,d) descending (Figure 9a,b) images at different time periods. Compared with the DEM error maps, it is easy to see that the spatial pattern of the DEM errors estimated by the first-stack images is inconsistent with that estimated by the second-stack images, indicating that the MECC projects were ongoing during the period of InSAR observation. For the SAR images with identical temporal spanning, the DEM errors estimated by the ascending images in spatial distribution and pattern were consistent with those estimated by the descending images. This cross-validates the accuracy and reliability of the estimated DEM errors in this study. The DEM error signals were very large in the study area, and the maximum DEM error was approximately 100 m for both ascending and descending SAR images. The estimated DEM errors were mainly distributed in the MECC areas around the main urban area of Lanzhou, and there was basically no DEM error (except for some high-rise construction areas) in the main urban area of Lanzhou (Figure 9).

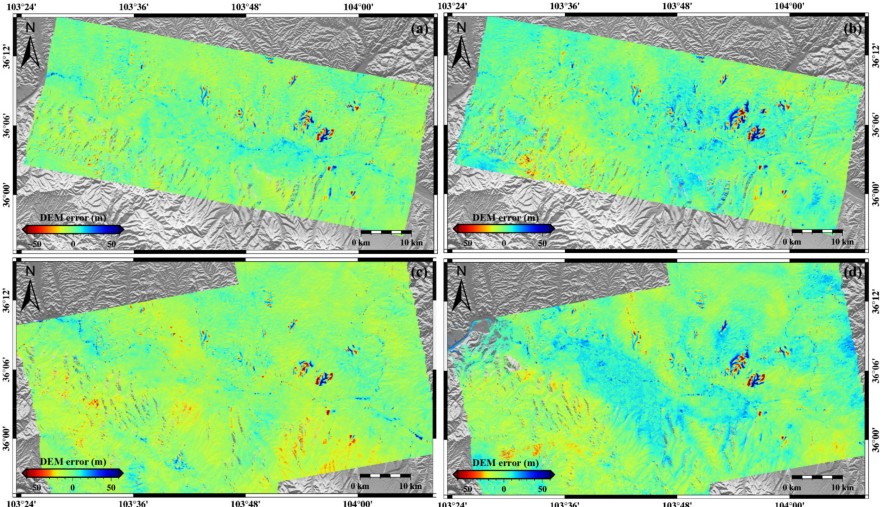

**Figure 9.** Estimated DEM errors of the study area at different time periods using ascending and descending Sentinel-1 SAR images: (**a**) DEM error between 8 December 2015 and 8 June 2020 estimated with descending images; (**b**) DEM errors between 20 June 2020 and 23 March 2021 estimated with descending images; (**c**) DEM error between 15 December 2015 and 16 June 2020 estimated with ascending images; and (**d**) DEM error between 2 August 2020 and 30 March 2021 estimated with ascending images.

Based on the estimated DEM errors, coupled with multi-temporal optical images, we detected and mapped historical MECC areas in Lanzhou during the period between 2000 and 2020 (Figure 10). We can see from Figure 10 that many lands (i.e., a total of 115 from 2000 to 2020) have been created in Lanzhou by the MECC projects accompanied with rapid urbanization. During the period of 2000 to 2005, the number of MECC area was nine; however, it has rapidly increased in the following 15 years. This number extended to 20, 36, and 50 during the periods of 2005–2010, 2010–2015, and 2015–2020, respectively. The larger land creation areas were identified in Regions A, B, C, and D and marked by white rectangles in Figure 10a. Figure 10b,d,f,h and Figure 10c,e,g,i show the optical images of Regions A, B, C, and D before and after creating the lands, respectively. It is evidently shown that these regions were covered by the mountains before creating the lands and were covered by dense buildings afterwards. Moreover, the terrains have been systematically remolded in these regions.

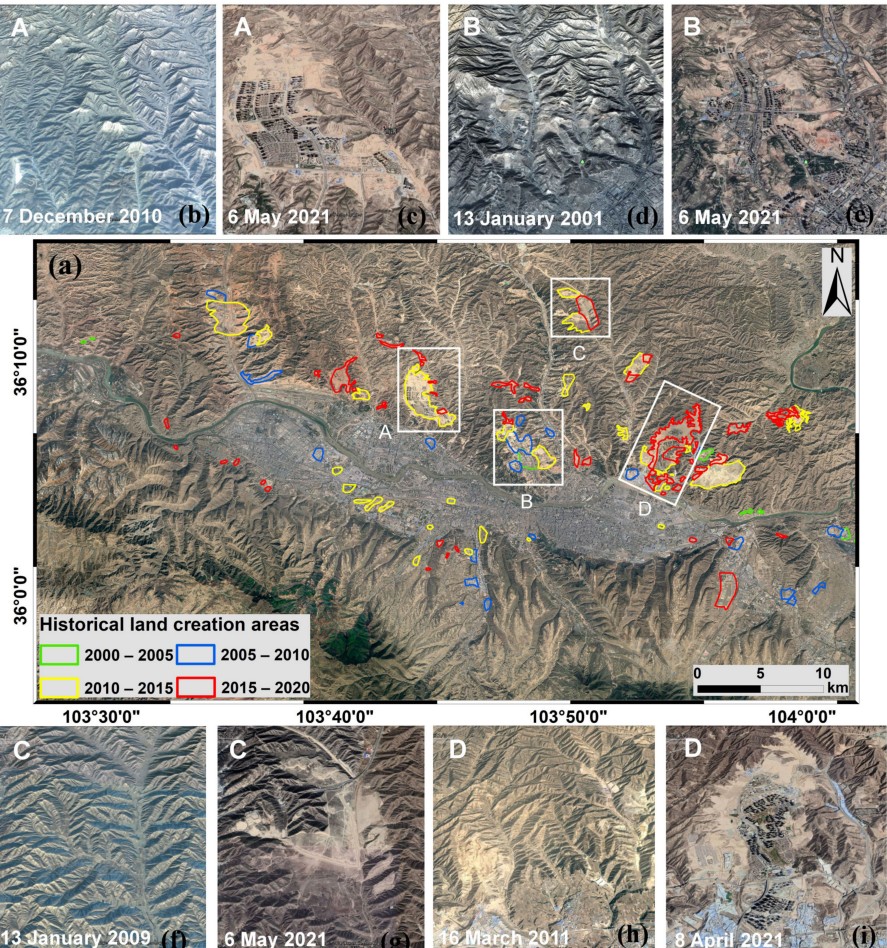

**Figure 10.** Spatial distribution and optical images of the historical MECC areas in Lanzhou detected by DEM errors: (**a**) spatial distribution of MECC areas in different time periods marked by different color polygons; (**b**,**c**) are the enlarged optical images of Region A acquired on 7 December 2010 and 6 May 2021, respectively; (**d**,**e**) are the enlarged optical images of Region B acquired on 13 October 2001 and 6 May 2021, respectively; (**f**,**g**) are the enlarged optical images of Region C acquired on 13 January 2009 and 6 May 2021, respectively; (**h**,**i**) are the enlarged optical images of Region D acquired on 16 March 2011 and 8 April 2021, respectively.

*4.3. Spatial Distribution of the Surface Displacement in Lanzhou*

Annual displacement rate maps of the city of Lanzhou between December 2015 and March 2021 were calculated using ascending and descending Sentinel-1 images (Figure 11). Figure 11a,c,e depicts the displacement rates derived from ascending images covering the

periods from 15 December 2015 to 28 December 2017, 28 December 2017 to 30 December 2019, and 30 December 2019 to 30 March 2021, respectively; and Figure 11b,d,f, shows the displacement rates derived from descending images covering the periods from 8 December 2015 to 21 December 2017, 21 December 2017 to 23 December 2019, and 23 December 2019 to 23 March 2021, respectively. Red colors (negative values) represent ground settlement and noise, and blue colors (positive values) represent ground uplift and noise. It can be found from Figure 11 that widespread surface displacements were observed by both ascending and descending Sentine-1 images in the study area, mainly ranging from −5 to −100 mm/year with a minimum value higher than −250 mm/year. A variety of factors contributed to surface displacement in the study area, including MECC projects, tunnel excavation, underground mining, and landslides among others; among which the large-scale MECC projects were the primary causative factor. The active displacement areas are mainly distributed on the north side of the study area, and six obvious settlement regions with a relatively large surface in Regions A, B, C, D, E, and F were identified. The activities of these regions were evidenced by optical images which suggested that these regions corresponded to the large-scale MECC areas. The statistics of the surface displacement in Regions A–F derived from ascending and descending Sentinel-1 images are presented in Tables 2 and 3, respectively.

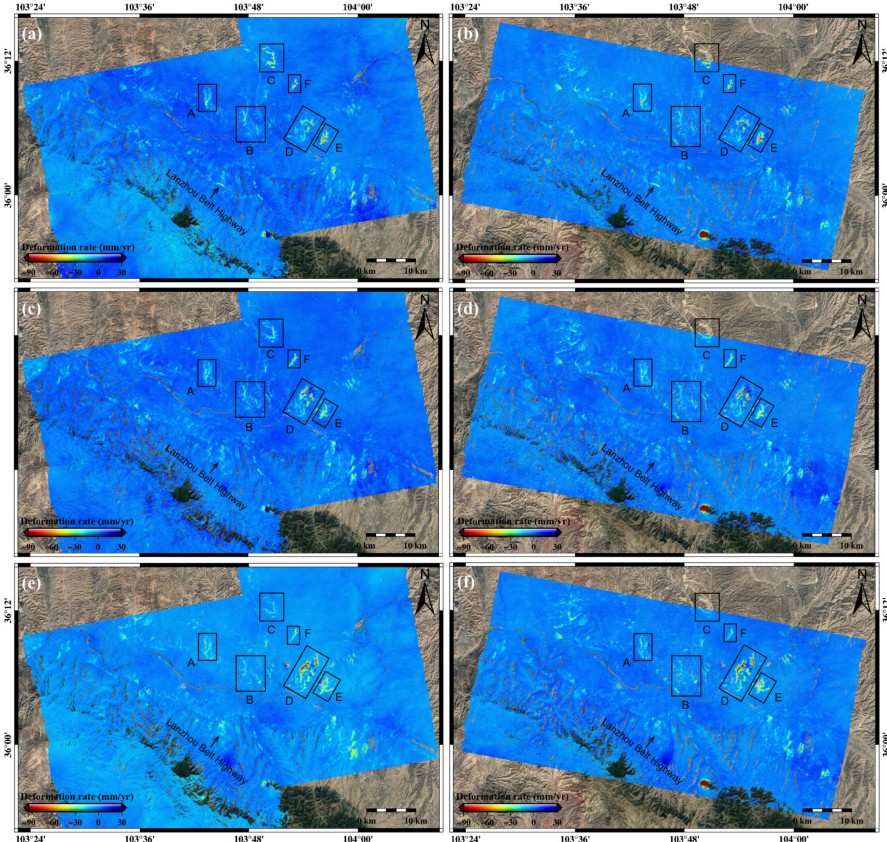

**Figure 11.** Annual displacement rates generated with ascending and descending Sentinel-1 images for the whole study area. The black rectangles (i.e., A, B, C, D, E and F) indicate six obvious land subsidence regions with relatively large area. (**a**) The displacement rate derived from ascending Sentinel-1 images between 15 December 2015 and 28 December 2017; (**b**) the displacement rate derived from descending Sentinel-1 images between 8 December 2015 and 21 December 2017; (**c**) the displacement rate derived from ascending Sentinel-1 images between 28 December 2017 and 30 December 2019; (**d**) the displacement rate derived from descending Sentinel-1 images between 21 December 2017 and 23 December 2019; (**e**) the displacement rate derived from ascending Sentinel-1 images between 30 December 2019 and 30 March 2021; and (**f**) the displacement rate derived from descending Sentinel-1 images between 23 December 2019 and 23 March 2021.

**Table 2.** Statistics of the surface displacement in Regions A–F, derived from ascending Sentinel-1 images.

| Region | | | A | B | C | D | E | F |
|---|---|---|---|---|---|---|---|---|
| Displacement rate (mm/year) | 2015.12–2017.12 | Min. | −5 | −5 | −5 | −5 | −10 | −8 |
| | | Max. | −70 | −55 | −70 | −80 | −90 | −86 |
| | | Ave. | −13 | −12 | −23 | −14 | −32 | −31 |
| | 2017.12–2019.12 | Min. | −5 | −5 | −5 | −5 | −10 | −8 |
| | | Max. | −56 | −46 | −36 | −94 | −90 | −88 |
| | | Ave. | −12 | −11 | −14 | −17 | −24 | −24 |
| | 2019.12–2021.03 | Min. | −5 | −5 | −5 | −15 | −10 | −8 |
| | | Max. | −38 | −55 | −32 | −123 | −101 | −46 |
| | | Ave. | −11 | −9 | −11 | −27 | −20 | −16 |
| Cumulative displacement (mm) | 2015.12–2021.03 | Min. | −20 | −20 | −20 | −27 | −30 | −30 |
| | | Max. | −233 | −200 | −223 | −409 | −372 | −338 |
| | | Ave. | −52 | −48 | −80 | −79 | −98 | −106 |

Note that, Min., Max., and Ave. represent the minimum, maximum, and average values of displacement, respectively.

**Table 3.** Statistics of the surface displacement in Regions A–F, derived from descending Sentinel-1 images.

| Region | | | A | B | C | D | E | F |
|---|---|---|---|---|---|---|---|---|
| Displacement rate (mm/year) | 2015.12–2017.12 | Min. | −5 | −5 | −5 | −5 | −10 | −8 |
| | | Max. | −76 | −49 | −72 | −72 | −94 | −86 |
| | | Ave. | −13 | −10 | −25 | −14 | −33 | −30 |
| | 2017.12–2019.12 | Min. | −5 | −5 | −5 | −5 | −10 | −8 |
| | | Max. | −66 | −45 | −40 | −96 | −96 | −85 |
| | | Ave. | −12 | −10 | −13 | −18 | −25 | −24 |
| | 2019.12–2021.03 | Min. | −5 | −5 | −5 | −12 | −10 | −8 |
| | | Max. | −38 | −53 | −27 | −118 | −88 | −40 |
| | | Ave. | −11 | −9 | −10 | −22 | −22 | −16 |
| Cumulative displacement (mm) | 2015.12–2021.03 | Min. | −23 | −20 | −20 | −27 | −30 | −31 |
| | | Max. | −239 | −186 | −215 | −421 | −365 | −344 |
| | | Ave. | −55 | −44 | −77 | −84 | −104 | −104 |

Note that, Min., Max., and Ave. represent the minimum, maximum, and average values of displacement, respectively.

As can be seen from Figure 11 and Tables 2 and 3, the displacement distributions, magnitudes, and patterns derived from ascending and descending images were very similar spatially, suggesting that the surface displacement in the study area was dominated by settlements, as will be demonstrated in Section 4.4. However, the displacement distributions, magnitudes, and patterns derived from both ascending and descending images were diverse temporally. The displacement rate decreased with time in most of the MECC areas, such as Regions A, B, C, and F. This displacement pattern has also been found in marine reclamation land [6] and is correlated to the consolidation function of the filling loess. For Regions D and E, some newly significant displacement areas were discovered during the period from December 2019 to March 2021, particularly in Region D. We can see from Figure 11c,d that the severe decorrelation signals occurred in the new displacement areas of Region D, indicating that the lands in these areas were created between December 2017 and December 2019. Besides, the displacement rate maps illustrate that non-uniform displacement exists in the MECC areas; this may be correlated with the thickness of the filling loess [4] and will be discussed in detail in Section 5.3. Compared with Figures 10 and 11, we found that most of MECC areas completed before 2005 were basically stable, and the main urban of Lanzhou was also stable. Moreover, a linear displacement area was identified in the south side of the study area. According to the optical images and the performed field investigation, this was caused by the excavation of the tunnels during the construction of the Lanzhou Belt Highway, which will be discussed in Section 4.5.

### 4.4. Two-Dimensional Patterns of Surface Displacement Associated with the MECC Projects

We selected Regions A and B marked in Figure 11 as the study sites to investigate two-dimensional (2D) surface displacement patterns associated with the MECC projects, because a great proportion of buildings had been constructed on these two regions during the studied period. The 2D displacement rates were calculated with the methodology described in Section 3.3, as shown in Figure 12. Red colors (negative values) in Figure 12a,c,e indicate downward movement, and red colors in Figure 12b,d,f represent westward movement. It can be clearly seen that the surface displacements in Regions A and B were dominated by the vertical movement, mainly ranging from −5 to −68 mm/year, −5 to

−78 mm/year, and −5 to −5 mm/year between December 2015 and December 2017, December 2017 and December 2019, and December 2019 and March 2021, respectively. These results suggested that surface displacements caused by the MECC projects were mainly manifested as settlement, and thus, horizontal displacement could be ignored in further processing. We can see from Figure 12 that Region A had a large area of settlements; however, the settlement areas were scattered in Region B; this was closely correlated with the completion time of the MECC project. In addition, in lands of Region B, there were no remarkable settlements created before 2010, and the velocity of the settlement in Regions A and B decreased with the time. For region A, evident settlements were distributed in a zone from north to south along the filling areas, and the largest displacement between December 2015 and December 2017 was detected at the south end of this settlement zone, and the one between December 2017 and December 2019 was detected at the middle part. For region B, the strong settlement mainly distributed at the northwest zones, which were created after 2010 according to Figure 10.

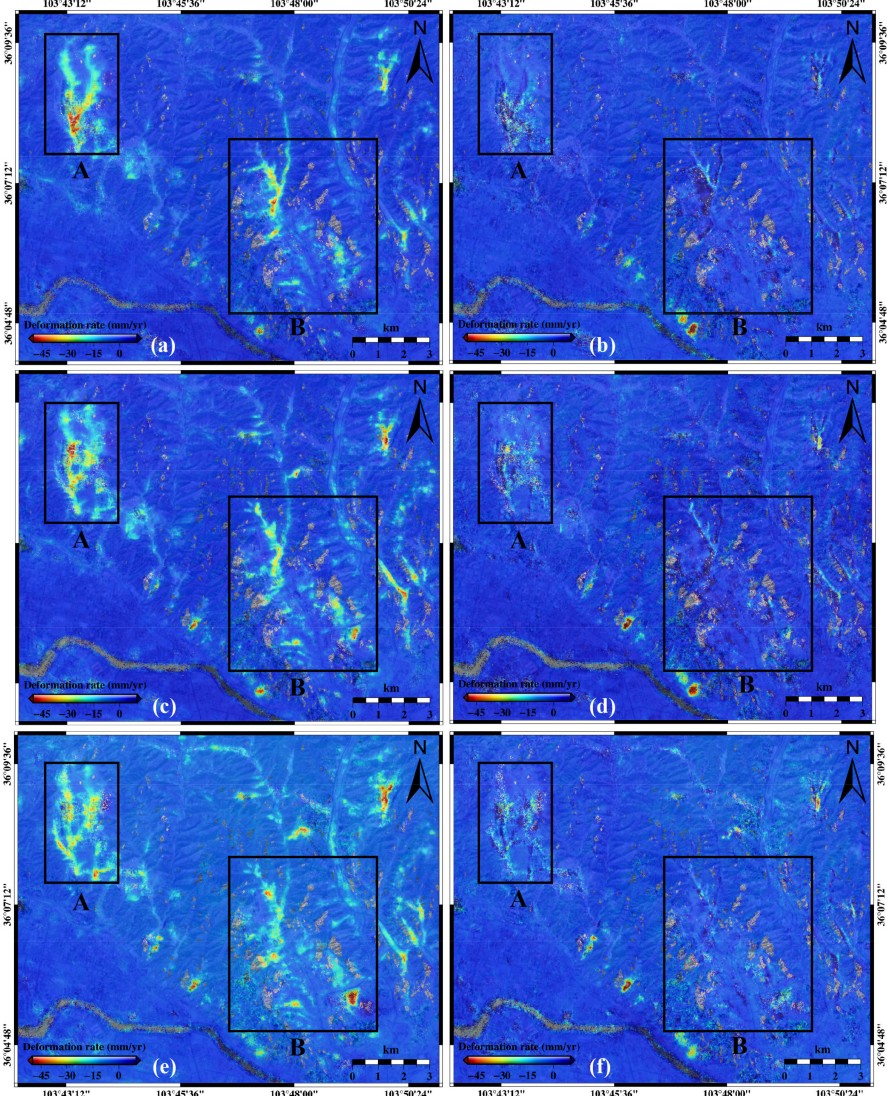

**Figure 12.** Two-dimensional displacement rates in the vertical (left) and east–west (right) directions of Regions A and B (marked in Figure 11) from December 2015 to March 2021, derived from ascending and descending Sentinel-1 images: (**a**,**b**) are the vertical and east–west displacement rates between December 2015 and December 2017, respectively; (**c**,**d**) are the vertical and east–west displacement rates between December 2017 and December 2019, respectively; (**e**,**f**) are the vertical and east–west displacement rates between December 2019 and March 2021, respectively.

### 4.5. Surface Displacement along the Lanzhou Belt Highway

As mentioned in Section 4.3, a linear displacement pattern was observed in the south side of the study area between 15 December 2015 and 28 December 2017. This displacement was caused by the excavation of tunnels during the construction of the Lanzhou Belt Highway (LBH), based on the evidence from optical images and field investigation. The construction of the LBH started in March 2015 and was mostly completed in October 2018. We enlarged the displacement rates along the LBH derived from ascending and descending Sentinel-1 images between December 2015 and December 2017, as shown in Figure 13. The active displacement areas detected by InSAR were all distributed in areas of the tunnel excavation, and the displacement rates ranging from −5 to −60 mm/year for both ascending and descending images. It can be seen that the magnitude and spatial distribution of displacements measured by ascending images were consistent with those measured by descending images, cross-validating the reliability of the estimated displacement results. This fact also suggested that the displacement generated by the excavation of the tunnels was dominated by the vertical movement. Additionally, non-uniform displacements were observed along the LBH; i.e., the displacement was only observed in some tunnel excavation areas; this might be closely related to the depth of the tunnel and its excavation method [44]. Three areas with larger displacement rates were detected along the LBH, that is, Regions F, G, and H, marked by black rectangles in Figure 13. The first area (i.e., Region F) was the Xijin tunnel, located in Xijin village, Qilihe district of Lanzhou, where many villages were placed on ground surface. The spatial displacement pattern of this area is enlarged in the upper right corner of Figure 13, in which we can see that the larger displacement was observed in the whole tunnel, with a maximum negative velocity of −53 mm/year. The second one (i.e., Region G) was adjacent to Chuandigang village, Qilihe district, and the other one (i.e., Region H) was situated in Dingyuan town, Yuzhon county, with a maximum negative velocity of −35 and −47 mm/year, respectively.

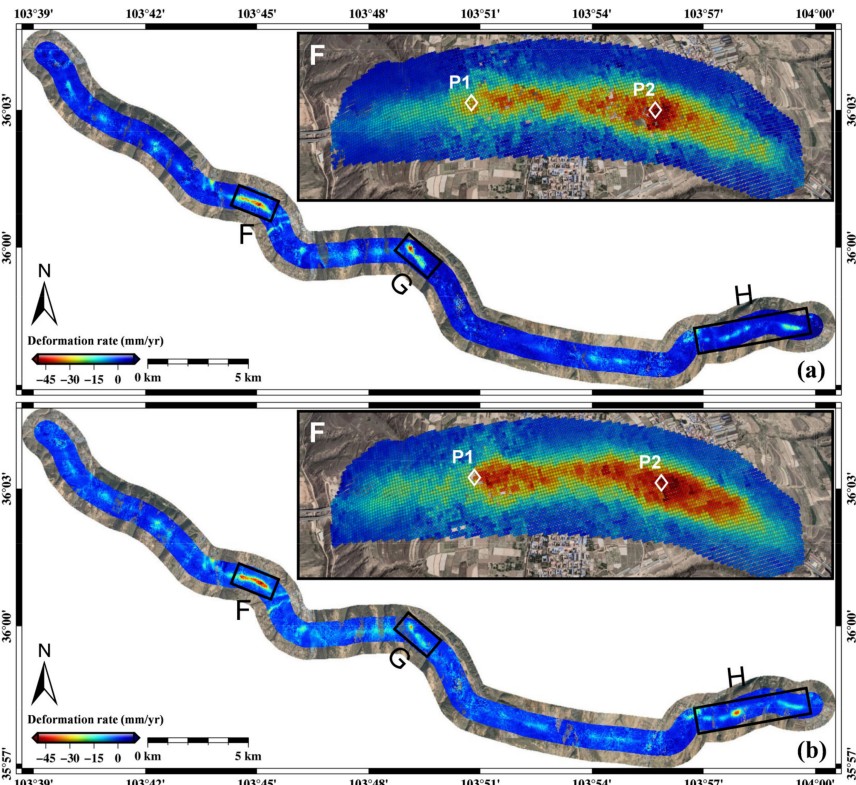

**Figure 13.** Annual displacement rates along the Lanzhou Belt Highway (LBH) from December 2015 to December 2017 calculated with ascending (**a**) and descending (**b**) Sentinel-1 images. Regions F, G and H, marked by black rectangles, are the three areas with larger displacement rates along the LBH, and Points P1 and P2 are used to show the displacement time series.

To validate our InSAR displacement results, a field investigation was performed at Region F (i.e., Xijin tunnel) in January 2021. Figure 14 shows four scene pictures taken from roads and buildings placed on the Xijin tunnel, where the black arrows indicate the cracks on the roads and buildings caused by the surface displacement. Evidence from field investigation demonstrated that the roads and buildings have been affected remarkably by the tunnel excavation, and they have been damaged to various degrees.

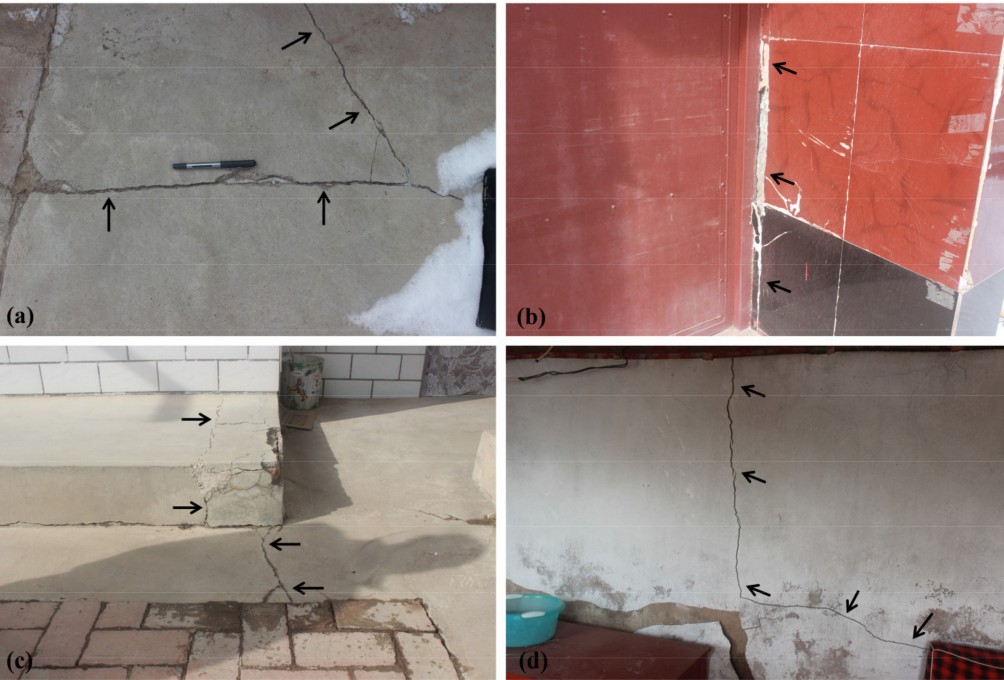

**Figure 14.** Field work photos showing some damage to roads and buildings caused by the tunnel excavation, where the black arrows indicate the cracks. (**a**) The damage to the road; (**b**) the damage to the building; (**c**) the damage to the road; and (**d**) the damage to the building.

Two representative points (i.e., Points P1 and P2 marked in Figure 13) were selected to characterize the temporal evolution behavior of the surface displacement related to the tunnel excavation. Figure 15 shows the time series of the displacement at Points P1 and P2 derived from ascending and descending Sentinel-1 images, spanning the period from December 2015 to March 2021. We can see that the displacement time series calculated with the ascending and descending Sentinel-1 images show a good consistency at each selected point, illustrating the reliability of the estimated displacement results. Evidently, non-linear displacement features were observed at Points P1 and P2. Ground surface was in a stable state before 13 April 2016 for Point P1 and before 2 November 2016 for Point P2, followed by a sudden acceleration displacement signal; this indicated that the tunnel excavations at Points P1 and P2 might have started on 13 April 2016 and 2 November 2016, respectively. The surface displacement developed progressively in accordance with tunnel excavation and advanced, and a progressive deceleration in displacement started to occur after the tunnelling completion on 18 September 2018. However, the entire area above the tunnels has not yet been completely stabilized by March 2021.

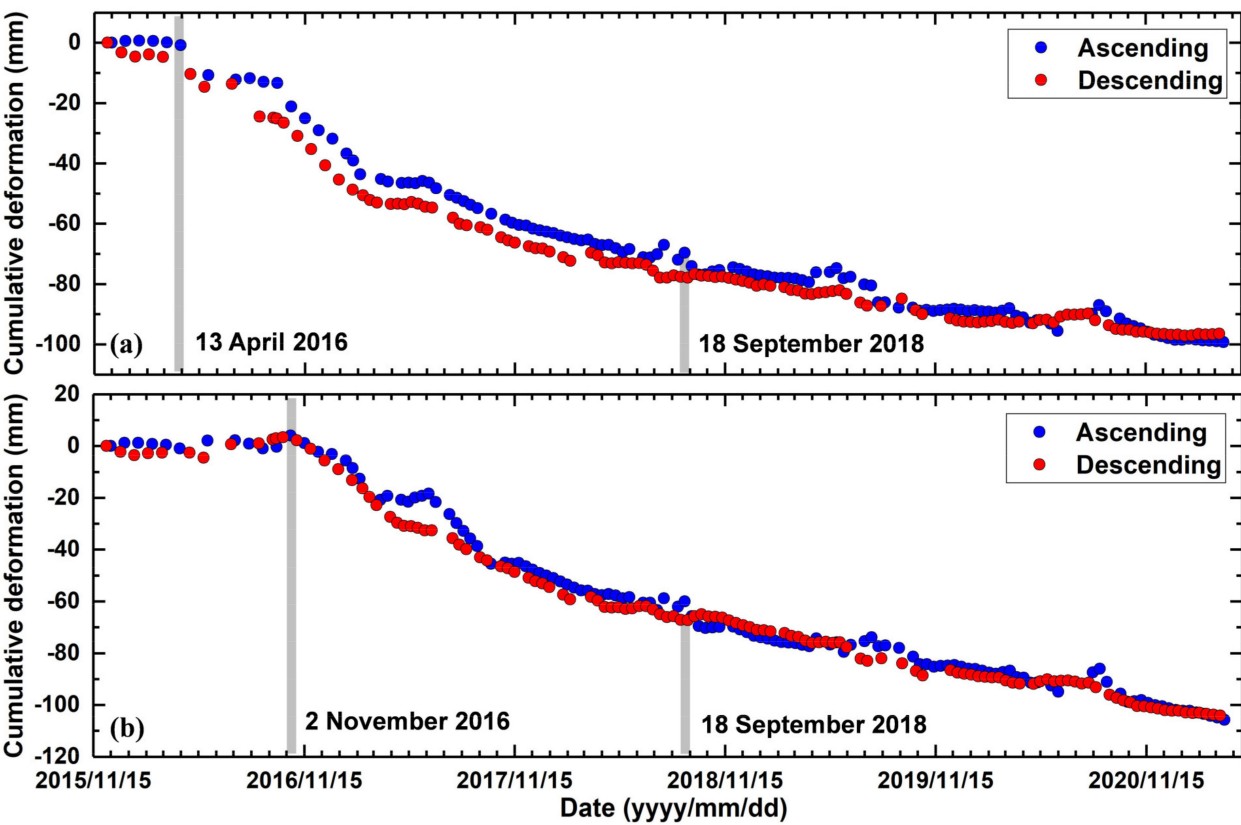

**Figure 15.** Time series of the displacement between December 2015 and March 2021 at Points P1 and P2 marked in Figure 13. The first grey bar indicates the time when the ground surface began to deform, and the second grey bar is the time when the tunnel excavation was completed. (**a**) P1; (**b**) P2. Back squares and red squares represent the ascending and descending displacements, respectively.

## 5. Discussion

### 5.1. The Impacts of Large-Scale Mountain Excavation and City Construction Projects on Surface Displacement

Displacement maps derived from ascending and descending Sentinel-1 images suggested that most of the active displacement areas in Lanzhou were distributed on the MECC areas. To quantitatively investigate the impact of the large-scale MECC projects on surface displacement, we detected and mapped the active displacement areas in Lanzhou by combining the displacement rates and time series derived from ascending and descending Sentinel-1 images and analyzed the correlation between the active displacement areas and the historical land creation areas (Figure 16). A total of 163 active displacement areas with diverse dimension were identified in the study area (black polygons in Figure 16). Clearly, there was an outstandingly high spatial correlation between active displacement areas and historical land creation areas. Quantitatively speaking, 110 detected active displacement areas were distributed at the MECC areas, others were caused by the landslides, tunnel excavation, and underground mining among others. This means that there exists a high spatial control of surface displacements in Lanzhou by large-scale mountain excavation and city construction projects. The filling materials in MECC areas came from the remodeling and backfilling of original loess, which are characterized by weak geomechanical properties and a loose structure [4]. As a consequence, the loess-filled foundations in the MECC areas are prone to compaction and consolidation under the coupled effects of self-gravity and external load, thus resulting in large surface displacements.

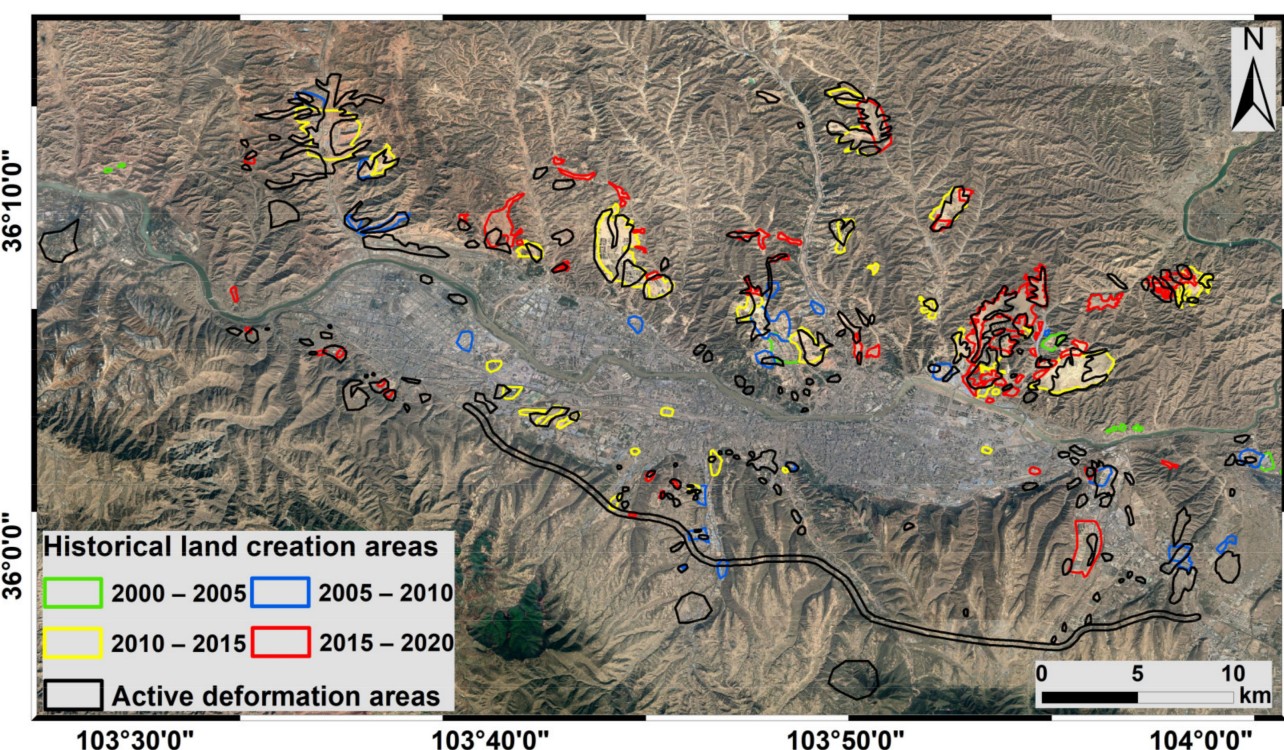

**Figure 16.** Correlation between active displacement areas and historical MECC areas in Lanzhou. Black polygons delineate active displacement areas in Lanzhou detected by InSAR-derived displacement maps.

### 5.2. Extraction of Excavation and Filling Areas and Its Volume Estimation

The obtained results can largely contribute to the analysis of surface displacement patterns to accurately identify the excavated and filling areas. However, there is no public bibliography to introduce the excavated and filling areas in the study area. Thus, we explored the use of DEM errors estimated by ascending and descending Sentinel-1 images to extract the excavated and filling areas. Region E marked in Figure 11 was selected as study site, the optical remote sensing images before and after land creation were collected, and the terrain elevation before and after land creation as well as the DEM errors were estimated with ascending and descending Sentinel-1 images (Figure 17). Figure 17a shows optical images acquired on 16 March 2011 (bottom) and 24 February 2021 (top), Figure 17b shows terrain elevation before (bottom) and after (top) land creation, and Figure 17c depicts the estimated DEM errors from ascending (top) and descending (bottom) Sentinel-1 images. In addition, the scene pictures of Region E are presented in Figure 17d–f. Moreover, terrain elevations and DEM errors were extracted along the Profile AA′ to clearly present topographic changes and excavated and filling areas (Figure 18). It can be found from Figure 17a that Region E was covered by mountains on 16 March 2011. However, it was modified on 24 February 2021, and the MECC project is currently ongoing (Figure 17d–f). As a result, we can see from Figures 17b and 18 that the topography in this region has been dramatically changed, and the maximum value in topographic change reached 95 m. According to the previous study [4], the maximum topographic changes in Yan'an, northwestern China caused by the MECC project was about 110 m. This suggests that the topographic change caused by the MECC project in Lanzhou is close to that in Yan'an.

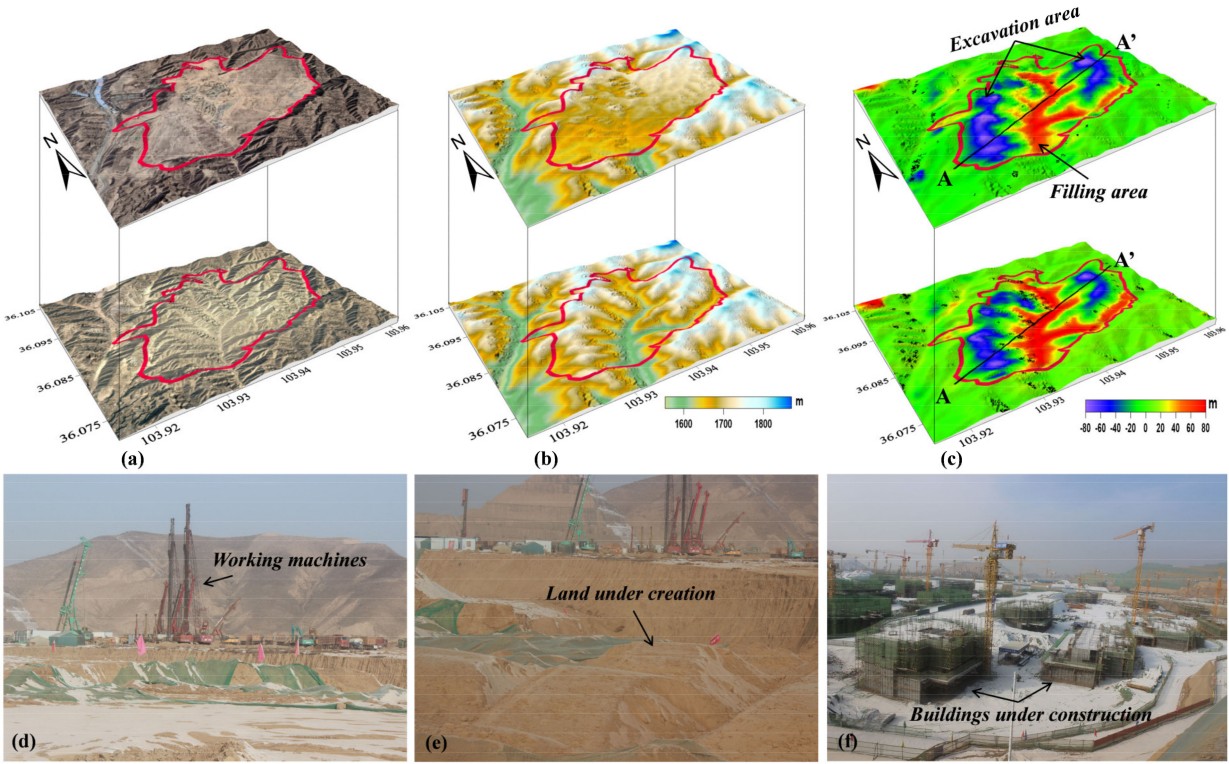

**(a)**        **(b)**        **(c)**

**(d)**        **(e)**        **(f)**

**Figure 17.** (**a**) Remote sensing images of Region E (see location in Figure 11) before (bottom) and after (top) creating land, which were acquired on 16 March 2011 (bottom) and 24 February 2021 (top), respectively. (**b**) Terrain elevation of Region E before (bottom) and after (top) creating land; (**c**) estimated DEM errors in Region E from ascending (top) and descending (bottom) Sentinel-1 images; (**d**–**f**) scene pictures of Region E taken in January 2021. Note that the red line indicates the boundary of the land creation in Region E.

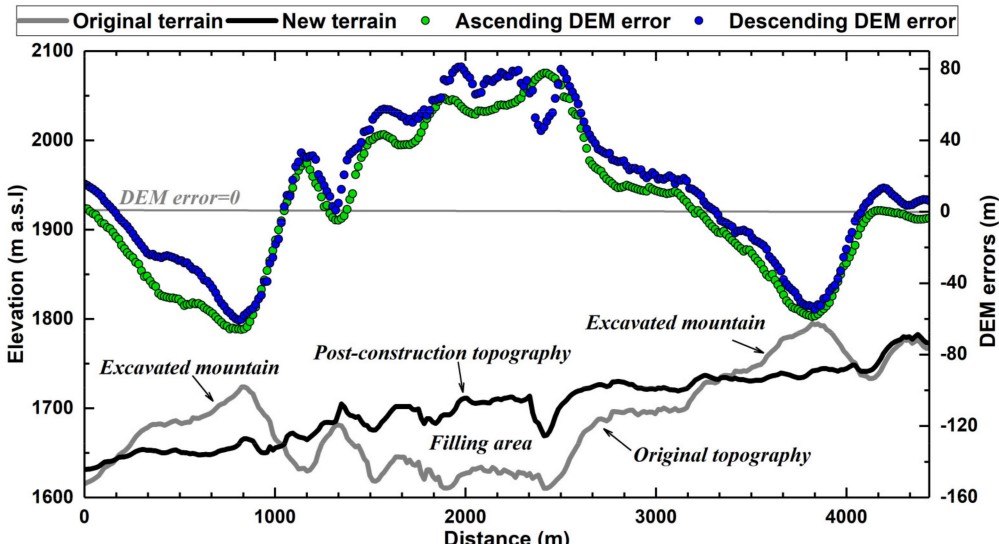

**Figure 18.** Comparison of terrain elevations before and after creating land and ascending and descending DEM errors along the Profile AA' marked in Figure 17c.

As can be seen from Figures 17c and 18, the DEM errors estimated by ascending images were highly consistent with those estimated by the descending images, demonstrating the reliability of the estimated DEM errors. The DEM errors evidently reveal the excavated and filling areas. Negative DEM errors were measured at both sides of Region E, which were located on the mountain areas of original topography (the grey line in Figure 18),

suggesting that these regions corresponded to excavated areas. In contrast, positive DEM errors were measured at the middle zone of Region E, which was located on the valley areas of original topography, suggesting that these regions were filling areas. Furthermore, we calculated the area and volume of the excavated and filling areas based on the estimated DEM errors. Results suggested that the areas of the excavated and filling areas were approximately 2.27 and 2.23 $km^2$, respectively, and the volumes of the excavated and filling areas were about 923.78 and 904.48 $km^3$, respectively. It can be concluded that the area and volume of the excavated area in Region E were basically equal to those of the filling area.

### 5.3. Surface Displacement Response to the Thickness of the Filling Loess

As mentioned above, non-uniform surface displacements were observed in the study area by both ascending and descending images. Such a finding may be correlated with the thickness of the filling loess in the MECC areas, because a previous study [4] found that the distribution and thickness of the filling loess in the large-scale MECC projects can largely determine the distribution and magnitude of the surface displacement. This relationship between soil thickness and settlements is in agreement with the soil mechanics theory. We estimated the thickness of the filling loess in the study area based on the InSAR-derived DEM errors. Regions A and E (marked in Figure 11) were selected as study sites to investigate the response of the surface displacement to the thickness of the filling loess. Figure 19a,b show the cumulative displacement from December 2015 to March 2021 and the DEM errors in Region E, respectively, and Figure 19c,d show the cumulative displacement and the DEM errors in Region A, respectively. The positive DEM errors in Figure 19b,d indicate the thickness of the filling loess. In addition, the cumulative displacement and the DEM errors were extracted along Profiles BB′ and CC′ to investigate the spatial distribution and pattern of surface displacement in depth, as shown in Figure 19e,f. Moreover, we conducted a correlation analysis to quantitatively identify the relationship between the surface displacement and the filling thickness. To this aim, the cumulative displacement and the filling thickness at each pixel were extracted and then correlatively analyzed (Figure 20).

Evidently, it can be found from Figure 19 that surface displacements were all distributed in the filling areas, and there was no remarkable surface displacement in the excavated areas. Such a displacement pattern can be easily explained in the MECC projects. The excavated areas are covered by the original loess strata, which have naturally developed through a period longer than 5000 years and experienced a series of complex process including compression, collapsible displacement, consolidation, and compaction [45]. Thus, the excavated areas are in a stable state. In contrast, the filling areas are covered by the remolded loess stratum, which are characterized by loose arrangement of soil particles, weak structure, and poor engineering features [23]. Although the foundations of the filling areas were consolidated using dynamic compaction method during the land creation process, it could enhance the strength of the filling loess only to some extent; however, it could not alter the physical characteristics of the collapsible loess. Consequently, the foundation of filling loess is not tamped completely, and it is more prone to deform under the coupled effects of self-gravity and external load.

As can be seen from Figure 19e,f and Figure 20, there is a certain correlation between the accumulative surface displacement and the thickness of the filling loess; i.e., in areas with large filling thickness, a significant surface displacement can be observed. The largest filling thicknesses in Regions A and E were 39.8 and 90.0 m, respectively, and the maximum cumulative displacements in Regions A and E between December 2015 and March 2021 were −239.0 and −364.8 mm, respectively. It is worth to specify that, to better visualize cumulative displacement results, the color range was set as between −300 and 20 mm in Figure 19a and between −150 and 20 mm in Figure 19c. The correlation coefficients (R) between cumulative displacement and filling thickness in Regions A and E were 0.36 and 0.63, respectively, and the linear fitting equations between them are presented in Figure 20. The cumulative displacement in Region E was much larger than that in

Region A because the filling thickness in Region A was smaller than that in Region E. The maximum cumulative displacement along the Profile BB' was measured at the middle of the profile, where the largest thickness of the filling loess was observed simultaneously (Figure 19e). A similar displacement pattern can also be observed along the Profile CC'. It is worth to specify that the correlation coefficient between cumulative displacement and filling thickness in Region A was relatively small, this may be due to the coupling effects of filling thickness and building load on surface displacement [2]. These outcomings demonstrate that, according to soil mechanics theory, there is a positive correlation between the magnitude of the surface displacement and the thickness of the filling loess; namely, the cumulative surface displacement increases with the thickness of the filling loess. These findings indicate that the compaction and consolidation process of the filling loess after MECC projects largely dominate the surface displacement.

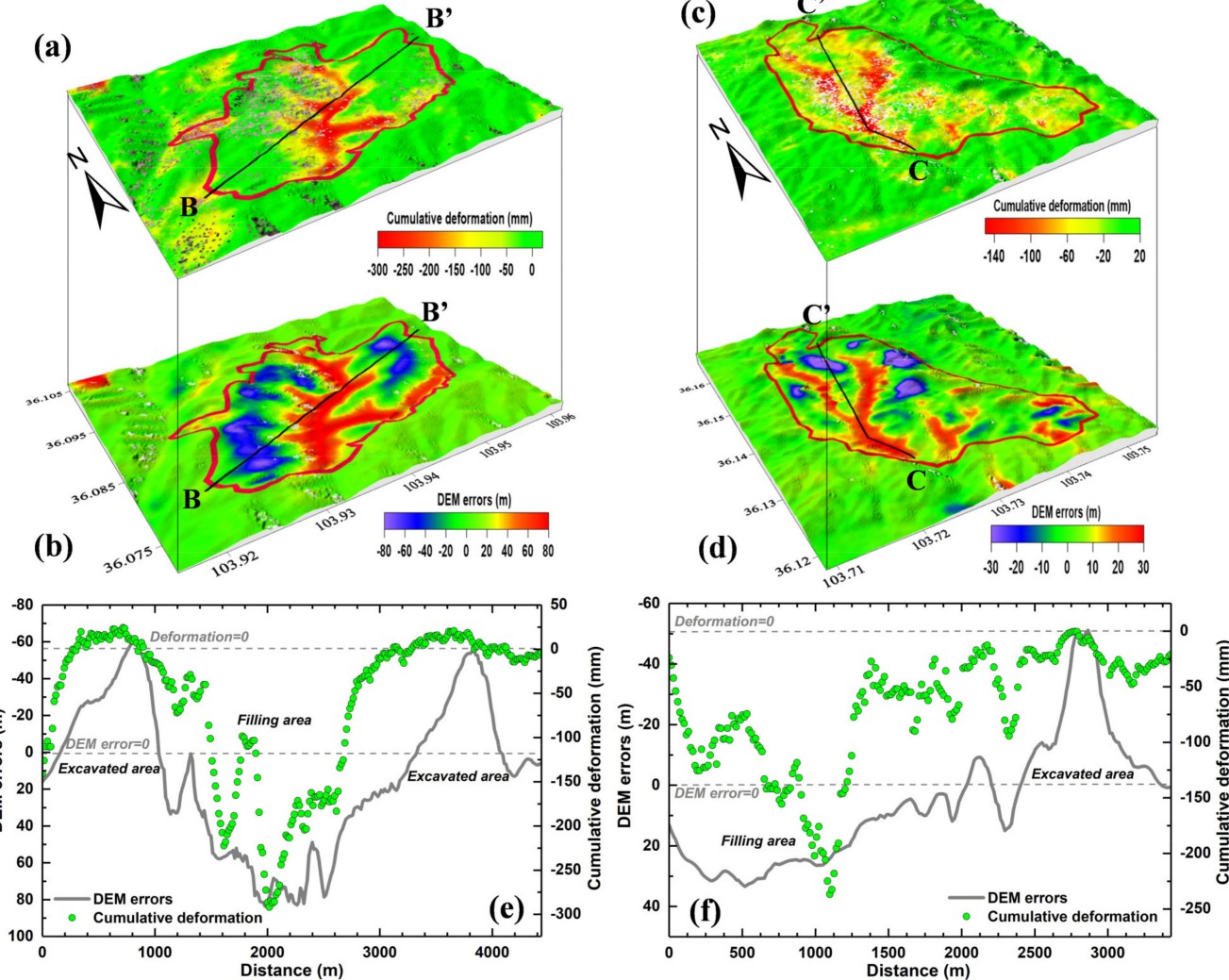

**Figure 19.** Relationship between surface displacements and thickness of the filling loess: (**a**,**b**) are the InSAR-derived cumulative displacements of Region E between December 2015 and March 2021 and the estimated DEM errors, respectively; (**c**,**d**) are the cumulative displacements of Region A between December 2015 and March 2021 and the estimated DEM errors, respectively; (**e**,**f**) are the cumulative displacements and estimated DEM errors along Profiles BB' and CC', respectively. The locations of Regions A and E are marked in Figure 11, and the locations of Profile BB' and CC' are presented in Subfigures a and c. Positive DEM errors in Subfigures b and d can be regarded as the thickness of the filling loess.

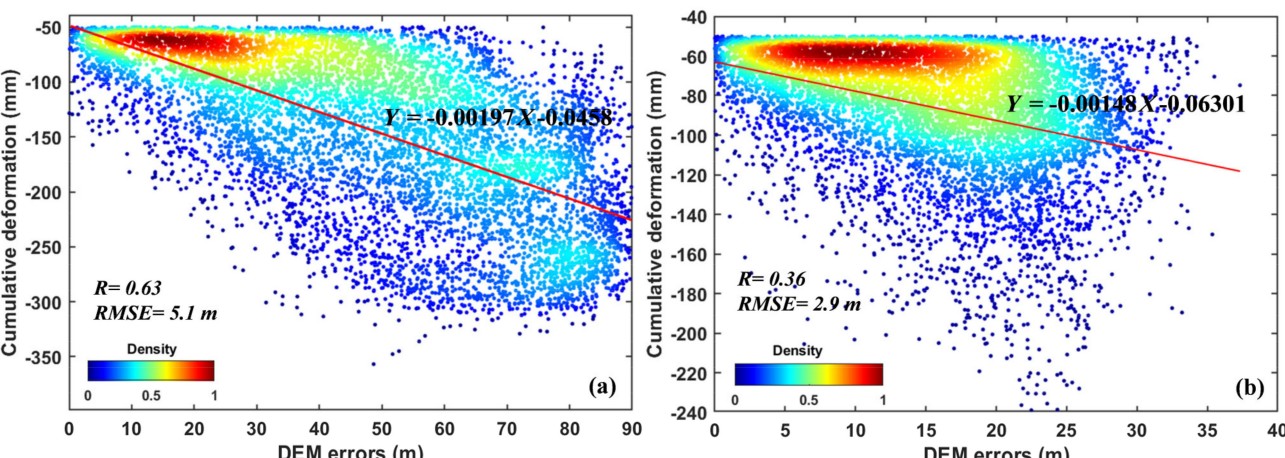

**Figure 20.** Correlation analyses between surface displacements and thickness of the filling loess in Regions E (**a**) and A (**b**). The red lines indicate the linear fitting between cumulative displacement and DEM errors (i.e., filling thickness), R is the correlation coefficient, and RMSE stands for root mean square error.

*5.4. Temporal Evolution of the Surface Displacement Associated with the MECC Projects*

Regions D and F marked in Figure 11 were selected as study sites to investigate temporal evolution behaviors of the surface displacement associated with MECC projects. Figures 21 and 22 show the cumulative displacement maps of Regions D and F from December 2015 to March 2021 derived from descending Sentinel-1 images, respectively. In addition, the settlement areas with larger cumulative displacement are highlighted in Figures 21b–d and 22b,c. The MECC projects were still ongoing in Regions D and F during InSAR monitoring period; thus, we used the cumulative displacement instead of the annual displacement rate to clearly reveal the displacement patters. Furthermore, eight exemplary points (Points P3–P10) located on different areas were selected to show the time series of the surface displacement, where Points P3–P6 were located in the Region D, and Points P7–P10 were situated in the Region F. Displacement time series for Points P3–P6 and P7–P10 between December 2015 and March 2021 are presented in Figures 23 and 24, respectively.

From Figure 21, we can clearly see that the cumulative surface displacement in Region D was significantly large, with a maximum value of −42.1 cm from December 2015 to March 2021. Nine remarkable land subsidence zones exist in the filling areas of Region D. Compared with Region D, the cumulative displacement in Region F was relatively small (Figure 22), with a maximum negative value of −34.4 cm; this can be attributed to two reasons. On the one hand, Region F was constructed earlier than Region D; on the other hand, the thickness of the filling loess in Region F was smaller than in Region D (Figure 9). As can be seen from Figures 23 and 24, the displacements of Points P3–P10 exhibited a non-linear trend with diverse velocities. For Points P4, P7, and P9, the temporal evolution of the displacement can be divided into two stages: rapid displacement stage and slow displacement stage. For instance, Point P4 was in a rapid displacement stage before 30 July 2017, and then it begun a slow displacement period between 20 July 2017 and 23 March 2021 (Figure 23b). Such a temporal evolution behavior is in agreement with the displacement pattern predicted by the consolidation theory of unsaturated soils [6]. Therefore, we can conclude that the internal mechanism of land subsidence at Points P4, P7, and P9 is the consolidation and compaction displacement of the filling unsaturated loess. For Points P3, P5, P6, P8, and P10, the temporal evolution of the displacement can also be divided into two stages: stable stage and rapid displacement stage. For example, the Point P3 was in stable state before 13 August 2019, followed by a sharply accelerated displacement between 13 August 2019 and 23 March 2021, indicating the MECC project at Point P3 was launched on 13 August 2019. Such a sharply accelerated displacement was mainly caused by the beginning of the consolidation of the filling unsaturated loess

that causes the simultaneous flow of air and water from the soil pores until equilibrium conditions were achieved [46].

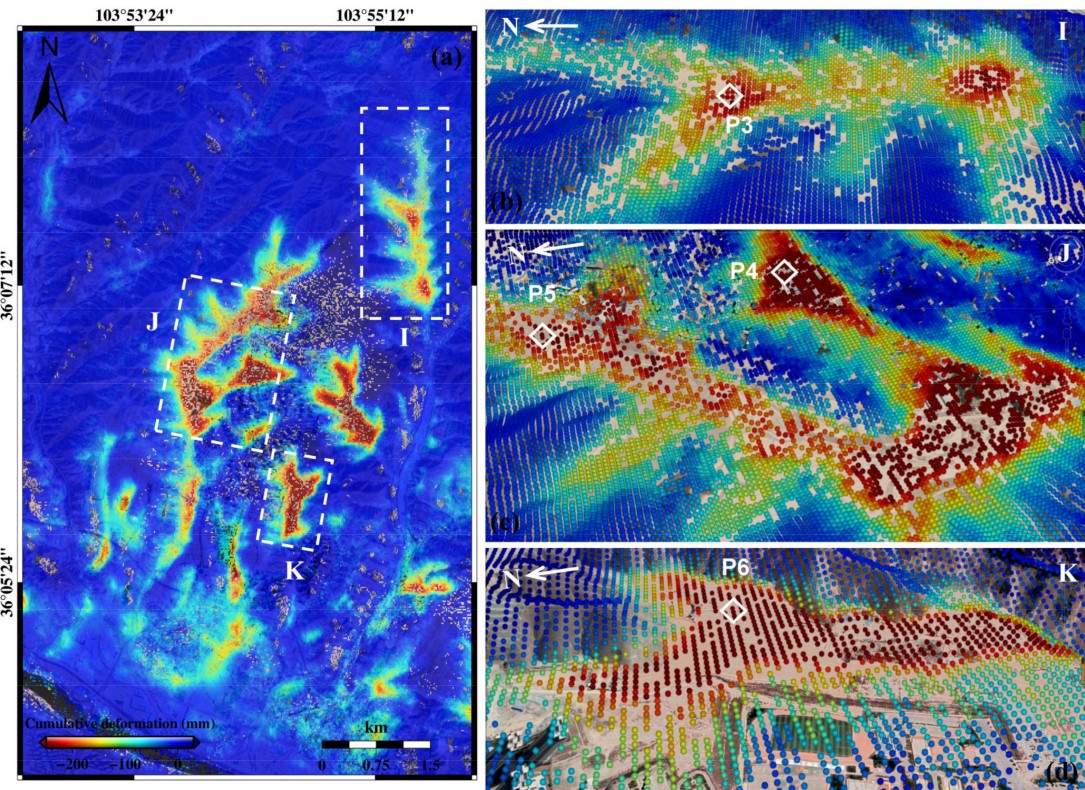

**Figure 21.** (**a**) Cumulative displacement map of Region D between December 2015 and March 2021. The location of Region D is marked in Figure 11. Regions I–K marked by white dashed rectangles are the three deforming areas with larger cumulative displacement, and their enlarged displacement maps are presented in (**b**–**d**), respectively.

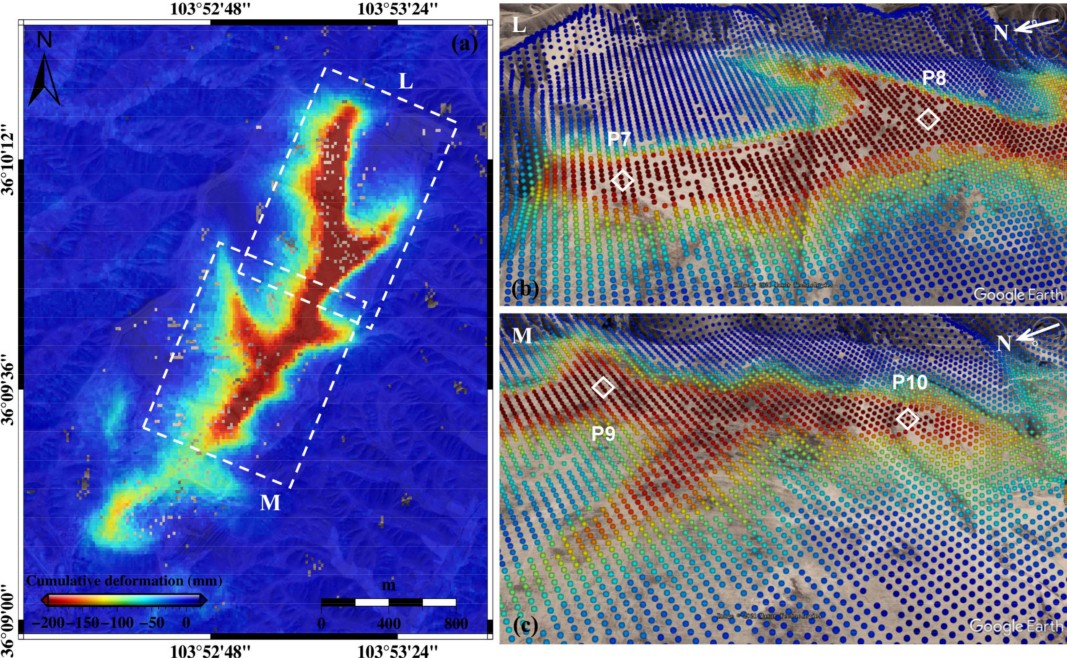

**Figure 22.** (**a**) Cumulative displacement map of Region F between December 2015 and March 2021. The location of Region F is marked in Figure 11. Regions L and M marked by white dashed rectangles are the two deforming areas with larger cumulative displacement, and their enlarged displacement maps are presented in (**b**,**c**), respectively.

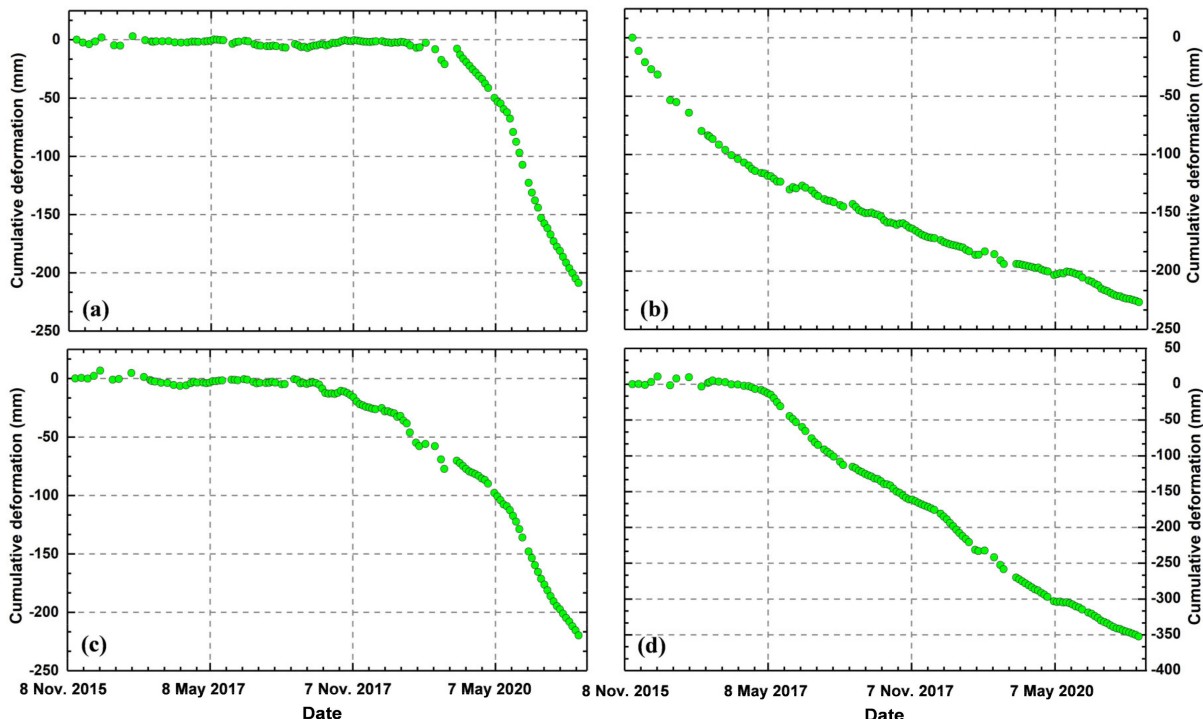

**Figure 23.** Time series of the surface displacement at Points P3–P6 from December 2015 and March 2021, the locations of Points P3–P6 are marked in Figure 21. (**a**) Displacement time series for Point P3 marked in Figure 21b; (**b**) displacement time series for Point P4 marked in Figure 21c; (**c**) displacement time series for Point P5 marked in Figure 21c; (**d**) displacement time series for Point P6 marked in Figure 21d.

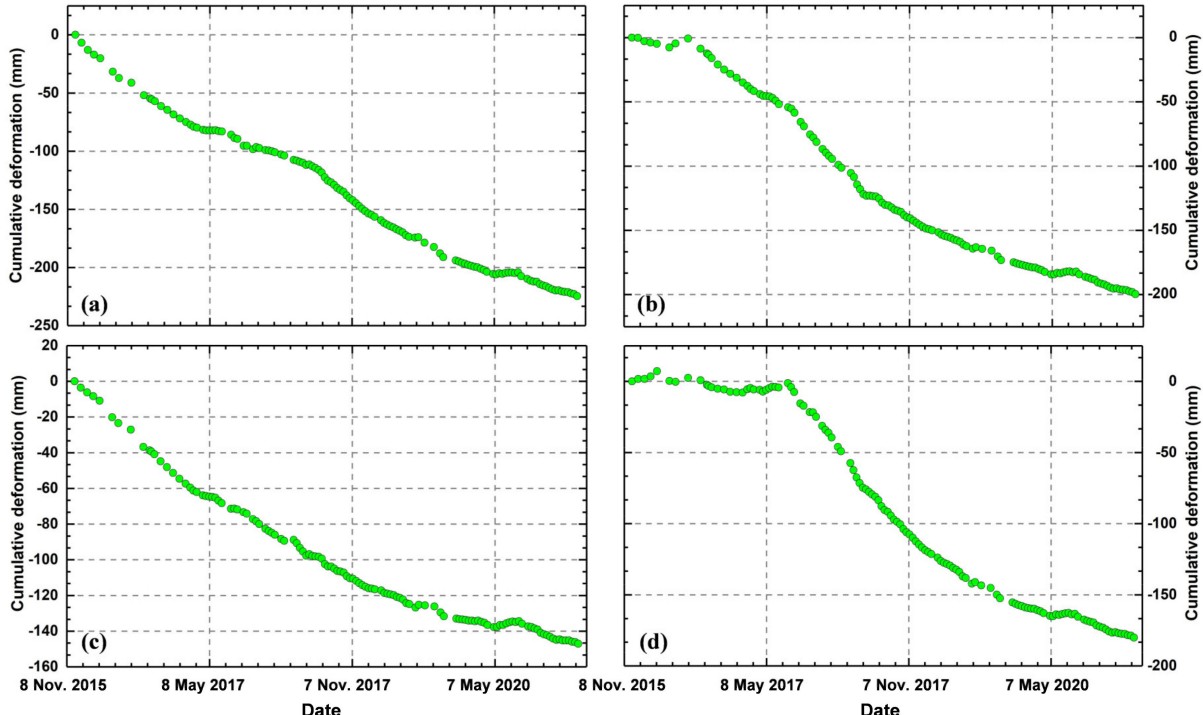

**Figure 24.** Time series of surface displacement at Points P7–P10 from December 2015 and March 2021. The locations of Points P7–P10 are marked in Figure 22. (**a**) Displacement time series for Point P7 marked in Figure 22b; (**b**) displacement time series for Point P8 marked in Figure 22b; (**c**) displacement time series for Point P9 marked in Figure 22c; (**d**) displacement time series for Point P10 marked in Figure 22c.

## 6. Conclusions

At present, there is no comprehensive displacement map characterizing the distribution and spatiotemporal patterns of surface displacement in the city of Lanzhou. In this work, we integrated satellite-based remote sensing data including multi-temporal optical images and multi-spaceborne SAR images of ascending and descending Sentinel-1 to investigate the distributions and spatiotemporal patterns of the surface displacement associated with the large-scale MECC projects in the study area. The optimized SAR processing procedure helps to retrieve high-precision surface displacement maps and dynamic DEM errors. Historical MECC areas and active displacement areas were detected and mapped using InSAR-derived DEM errors and surface displacement maps coupled with optical images. Beyond that, we also investigated the impacts of large-scale MECC projects on surface displacement and the response of surface displacements to the thickness of the filling loess and extracted the excavated and filling regions and their areas and volumes. The findings and outcomes generated in this study answered three concerns to some extent: (1) where the active displacement areas in Lanzhou are, (2) what the displacement mechanism is, and (3) how the temporal evolution behavior of the displacement is. The method and analysis presented in this study can be used for other cities worldwide with large-scale MECC projects. The conclusions and contributions of this study can be drawn as follows:

(1) DEM error is the main error source in the mapping of surface displacement associated large-scale MECC projects. This error can increase the difficulty of phase unwrapping and generate pseudo displacement signals. It should be carefully estimated and corrected in the SAR processing to avoid misinterpretations of the surface displacement signal.

(2) A total of 115 historical MECC areas were detected and mapped in the study area between 1997 and 2020, and 163 active displacement areas were identified with diverse dimensions. Of the detected active displacement areas, 67% were distributed in the MECC areas, what confirms that surface displacements in Lanzhou were mainly caused by large-scale MECC projects. Settlements were detected in the filling regions of the MECC projects following non-uniform spatial displacement patterns. By correlating displacement and filling thickness, we found that the magnitude of the cumulative displacement was positively correlated with the thickness of the filling loess. These findings demonstrate that large-scale MECC projects control the spatial distributions and patterns of surface displacement, and the filling thickness determines the final displacement magnitude.

(3) Results from estimated DEM errors showed that the area and volume of the excavated regions were basically consistent with that of the filling regions. We concluded that the amount of excavation and the amount of filling were in a balanced state. The displacement time series results revealed that ground surface in the study area deformed following a non-linear trend, and the velocity was distinct at different regions. Ground surface was in a stable state before land creations, followed by the sharply accelerated displacement with the beginning of the MECC project, and then the displacement was slowed down with the increasing time. This temporal evolution behavior of the surface displacement is in agreement with the displacement pattern predicted by the consolidation theory of unsaturated soil [46].

**Author Contributions:** Y.W. and X.L. performed the experiments, produced the results, and conducted field investigation. Y.W. and X.L. drafted the manuscript, and C.Z. and R.T. finalized the manuscript. R.T. and Z.J. contributed to the discussion of the results. All authors have read and agreed to the published version of the manuscript.

**Funding:** This research is funded by the Natural Science Foundation of China (Grant No. 41874005), the Natural Science Foundation in Gansu Province of China (Grant Nos. 1508RJZA094 and 20JR10RA180). This research was also supported by a Chinese Scholarship Council studentship awarded to Xiaojie Liu (Ref. 202006560031).

**Institutional Review Board Statement:** Not applicable.

**Informed Consent Statement:** Not applicable.

**Data Availability Statement:** The Sentinel-1 data used in this study were freely downloaded from Copernicus and ESA (https://scihub.copernicus.eu/dhus/#/home), accessed on 20 April 2021; and the one-arc-second SRTM DEM data is freely downloaded from the website https://e4ftl01.cr.usgs.gov/MEASURES/SRTMGL1.003/2000.02.11/, accessed on 20 April 2021.

**Acknowledgments:** The figures in this study were prepared using GMT and Matlab R2021a. We thank the editor and three anonymous reviewers for their insightful comments and suggestions to improve the quality of this article. We also thank the Google Earth Platform for providing the optical remote sensing images.

**Conflicts of Interest:** The authors declare no conflict of interest.

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
