# Peer review of "Observation of Surface Displacement Associated with Rapid Urbanization and Land Creation in Lanzhou, Loess Plateau of China with Sentinel-1 SAR Imagery"

_remotesensing, doi:10.3390/rs13173472_

Round 1
Reviewer 1 Report
Dear Authors
The study is based on both ascending and descending imagery; thus, greatly increases the trustability of the presented results.
Another interesting result is that DEM is a major source of errors. Owing to that SRTM is used for a billion applications, this conclusion is very significant and sets the demand for performing new studies, which would investigate the impact of different DEMs on various applications (including InSAR).
Nevertheless, some additional work is needed.
- Please change “Sentinel” to “Sentinel-1” on the title of the Article.
- Line 55. “Lanzhou New Distinct”. Should be “District”, I think.
- Section 2.2.1. The real source of the Optical imagery is needed. The resolution is claimed to be different in range and azimuth, which is unusual for optical systems. So, we are interested in what sensor was used. This is especially important, because if we can’t be sure in the propers co-registration of the optical images, we, as a result, cannot be sure that the results of DEM errors estimation phase are correct.
- The exact steps of interferogram preparation, as well as the used software, should be described.
- Please check for the double spaces.
- Methodology does not provide sufficient details. Each Phase should be described in detail. For example, the DEM errors estimation is not well described.
- While the graphical representation of the results is fascinating, some more quantitative information is needed. I would like to see a table with statistics of average/max/min subsidence numbers. Similar table, which would cover all the analyzed areas might be added to Supplements.
- The article states that Correlation analyses were done, but the numbers are not provided. For example, Figure 20 should show the equation, R2, and RMSE of the correlation.
Thank you!
Reviewer 2 Report
The manuscript titled "Observation of Surface Deformation Associated with Rapid Urbanization and Land Creation in Lanzhou, Loess Plateau of China with Sentinel SAR imagery”, Authors Yuming Wei, Xiaojie Liu, Chaoying Zhao, Roberto Tomás, Zhuo Jiang, describes the results of satellite-based observations, including optical remote sensing and synthetic aperture radar (SAR) measurements, to characterize the landscape and topography changes in Lanzhou (China) between 1997 and 2020, and investigate the spatio-temporal patterns of the surface deformation associated with the large-scale mountain excavation and city construction (MECC) projects from 2015 December to March 2021.
In my opinion, the manuscript describes an interesting approach to finding ground deformations using InSAR processing together with updated DEMs of an excavated area. The introduction and the state-of-the-art are well described and the text is well organized, with interesting insights and consistent results.
In the keywords I suggest to include “DEM errors” and “remote sensing images” which are essential in this study.
Some remarks/suggestions:
- Page 3, line 125: please, provide the meaning of all acronyms in the text (SRTM…) when are written for the first time;
- Page 5, line 186: “…Figures 1c…”. Figure 2?
- Page 6, line 220, table 1: in the text authors declare that the SAR images cover the period from October 2014 to April 2021 (lines 209-210), but in table 1 they declare “12/2015 – 04/2021”? What's the mistake?
- Page 6, line 231: typing errors;
- Page 12, line 426 “…in spatial distribution and pattern perfectly match with…”: please, moderate this term because it is based only on visual inspection… perfection needs correlation 1… is this the case?
- Page 14, Figure 10: please, improve the readability of the Dates (for example, in Figure 10b: 2010/12/07…);
- Page 15, line 508: “…pattterns…”;
- Page 15, line 521: “…Region R…”;
- Page 16, line 533: “…Sentine-1…”;
- Page 19, Figure 15: please, improve the readability of the Dates (for example, in Figure 15a: 2016/04/13…);
- Page 21, lines 633-634: “…in Figures 6(d)-(f).”. Please check;
- Page 21, Figure 17: please, enlarge the coordinate labels of Figures (b) and (c) to improve readability;
- Page 22, line 652: “…along profile AA’ marked in Figure 11(c).”. Please correct;
- Page 22: please, throughout the manuscript insert the Figures in the text “close” the citation: Figures 19 and 20 are mentioned on page 22, but are presented on page 24, making difficult the readability. Similarly, Figure 10 is mentioned on page 12 but presented on page 14: if possible, reduce the space between the citation and the presentation of the Figures;
- Page 23, line 706: “…and -364.8 mm…”: please, check this data in the Figure 19;
- Page 25, Figure 21: Figures (c) and (d) are deformed compared to Figure (a): please, select different parts of Figure (a), but do not deform them;
- Page 26, line 771: “…Figure 21. (a) P3; (b) P4; (c) P5; (d) P6”. P3 is in (b), P4 in (c)… please check;
- Page 26, line 763: I think is P3;
- Page 27, line 778: “…are marked in Figure 23. (a) P7; (b) P8; (c) P9; (d) P10.”. Please check from Figure 23;
- Page 28, line 819: “…perfectly…”: please, moderate this term (see above).
Reviewer 3 Report
The main subject of the reviewed manuscript is the topography changes in the Lanzhou city on China’s Loess Plateau. There are many mountain excavation areas around the city. Many remote sensing technologies have been jointly introduced for the topography characterizing for the time 2017-2020. After that, the Sentinel-1 InSAR images were analyzed to detect the ground movement in the time 2015-2021.
The abstract shows the main goals of the research clearly and concisely. The first suggestion for the abstract would be to clarify more precisely the meaning of a “…optimized SAR processing procedure”. The areas of mining and deposition were further analyzed from the volume changes point of view. Authors find out the correlation between mined out and filled out volumes. They stated that the deformation is positive correlated with the thickness of the filling loess. It seems to be obvious because of the natural compaction of the loess. Is not it ?
The methodology was clearly and properly presented. The results as well as discussion bases directly on the presented method. Results proved the introduced methodology.
The interesting issue of the research was the improvement of DEM model according to the changes in the topography. This process was involved in the research technology. Although the presented methods are not novel in the meaning of the state of the art, the presented methodology is concise and shows well the possible applications of InSAR together with mathematical modeling. The manuscript presents the research for the interesting case study of the developing city area. The cities like Lanzhou are in a difficult situation because of their mountainous location.
The reviewer has a linguistic problem with the word “deformation” used in the manuscript. The physical meaning of this term relates to a strain and further with a strain tensor. In the manuscript, the displacements in different directions were discussed, rather than deformation. Please convince the reviewer about their hesitation in this issue.
Authors cited the research done in the same area of knowledge, they could introduce more novel papers and give better international background for the presented research. The manuscript has been written well and could attract an audience to the journal.
